# Learning Equivariances and Partial Equivariances from Data

## Abstract

Group equivariant Convolutional Neural Networks (G-CNNs) constrain features to respect the chosen symmetries, and lead to better generalization when these symmetries appear in the data. However, if the chosen symmetries are not present, group equivariant architectures lead to overly constrained models and worse performance. Frequently, the distribution of the data can be better represented by a subset of a group than by the group as a whole, e.g., rotations in $[-90°, 90°]$. In such cases, a model that respects equivariance *partially* is better suited to represent the data. Moreover, relevant symmetries may differ for low and high-level features, e.g., edge orientations in a face, and face poses relative to the camera. As a result, the optimal level of equivariance may differ per layer. In this work, we introduce *Partial G-CNNs*: a family of equivariant networks able to learn partial and full equivariances from data at every layer end-to-end. Partial G-CNNs retain full equivariance whenever beneficial, e.g., for rotated MNIST, but are able to restrict it whenever it becomes harmful, e.g., for 6 / 9 or natural image classification. Partial G-CNNs perform on par with G-CNNs when full equivariance is necessary, and outperform them otherwise. Our method is applicable to discrete groups, continuous groups and combinations thereof.

## 1 Introduction

The translation equivariance of Convolutional Neural Networks (CNNs) (LeCun et al., 1998) has proven to be an important inductive bias for good generalization on vision tasks. This is achieved by restricting learned representations to respect the translation symmetry observed in visual data, such that if an input is translated, its feature representation is also translated, but not modified in any other way. Group equivariant Convolutional Neural Networks (G-CNNs) (Cohen & Welling, 2016) extend equivariance to other symmetry groups. Analogously, they restrict the learned representations to respect the symmetries in the group considered such that if an input is transformed by an element in the group, e.g., a rotation, its feature representation is also transformed, e.g., rotated, but not modified.

However, the group to which G-CNNs are equivariant must be fixed prior to training, and imposing equivariance to a group larger than the symmetries present in the data leads to over-constraining and worse performance (Chen et al., 2020). The latter results from a difference in the distribution of the data, and the family of distributions the model can describe. Consequently, the group must be selected carefully, and it should correspond to the transformations that appear naturally in the data.

Interestingly, the distribution of the data can frequently be better represented by a subset of the group than by the group itself, e.g., rotations in $[-90°, 90°]$. For instance, natural images are much more likely to show an elephant standing straight or slightly rotated than an elephant upside-down. In some cases, group transformations can even change the desired model response. A typical example is the classification of the numbers 6 and 9, whose defining factor is their pose. In both examples, the distribution of the data is better represented by a model that respects rotation equivariance *partially*. That is, a model equivariant to some, but not all rotations.

In addition, the optimal family of full / partial equivariances may change per layer. This results from changes in pose likelihoods for low and high-level features in the data, e.g., the orientations of edges in an human face, and the pose of human faces relative to the camera. Whereas detecting a face may benefit from partial rotation equivariance, detecting edges benefits from full rotation equivariance.

These observations indicate that constructing a model with different levels of partial / full equivariances at each layer may be advantageous. Weiler & Cesa (2019) empirically show that manually selecting the level of equivariance at different layers can lead to accuracy improvements. However, manually defining layer-wise levels of equivariance is at best a difficult and time-consuming task.

In this work, we introduce *Partial Group equivariant Convolutional Neural Networks*: a new family of equivariant methods able to *learn* partial and full equivariances directly from data at every layer. This is achieved by defining a learnable probability distribution over group elements at each group convolutional layer in the network. Instead of sampling group elements uniformly from the group during the group convolution – as normally done –, Partial G-CNNs sample them from the learned distribution. This allows them to adjust their level of equivariance during training. We note that our partial equivariance framework allows for both equivariance forgetting, and full equivariance. The former is obtained by collapsing the distribution along a certain group dimension, e.g., the rotation axis, and the later results from keeping the learned distribution uniform over the entire group.

We evaluate Partial G-CNNs on illustrative toy tasks and on challenging benchmark datasets. We show that whenever full equivariance is beneficial, e.g., for rotated MNIST, Partial G-CNNs remain fully equivariant. However, whenever harmful, our models restrict equivariance to a subset of the group transformations, e.g., to make 6 / 9 classification possible or to improve on natural image classification. Our method improves upon conventional group equivariant networks when equivariance reductions are advantageous, and matches their performance whenever their design is optimal. To the best of our knowledge, this work is the first approach to explore partial group equivariances, and to provide a practical implementation to learn them directly from data in an end-to-end manner.

In summary, our **contributions** are:

- We present a novel architectural design for equivariant networks, with which group convolutional networks are able to learn layer-wise partial or full equivariances from data.

- We empirically show that partial equivariance allows our networks to improve upon conventional G-CNNs for tasks for which full equivariance is harmful. However, whenever beneficial, our models learn to stay fully equivariant and match the performance of G-CNNs.

## 2    RELATED WORK

**Equivariance learning.** Learning equivariant mappings from data has been explored by (*i*) meta-learning as a means to iterate between the learning of free weights and a weight-tying matrix that encodes the symmetry equivariances of the network (Zhou et al., 2020), or by (*ii*) learning the Lie algebra generators of the group jointly with the parameters of the network (Dehmamy et al., 2021). However, both approaches utilize the same learned symmetries across layers. In addition, MSR (Zhou et al., 2020) is only applicable to (small) discrete groups, and requires longer training times. L-Conv (Dehmamy et al., 2021), on the other hand, is only applicable to continuous groups and is not fully compatible with current deep learning components, e.g., pooling, normalization.

Contrary to these approaches, our method learns different levels of full/partial equivariance at every layer, is fully compatible with current deep learning components, and is applicable for discrete groups, continuous groups and combinations thereof. It is important to mention, however, that Zhou et al. (2020); Dehmamy et al. (2021) aim to learn the structure of the group from scratch. Our method, on the other hand, starts from a (very) large group and allows layers in the network to constrain their equivariance levels whenever necessary in order to fit the distribution of the data better.

**Invariance learning.** Learning the right amount of invariance from data has been explored by (*i*) learning a probability distribution over data augmentations, e.g., rotation, scaling, and passing the responses of the network to the augmented inputs through a voting mechanism (Benton et al., 2020), or by (*ii*) minimizing the lower bound on the marginal likelihood of the problem to infer the right amount of rotation invariance in the weights of a network (van der Ouderaa & van der Wilk, 2021).

Contrary to these approaches, our method learns full / partial equivariant representations, and is able to modify the extent of equivariance at every layer. Most similar to our approach is Augerino (Benton et al., 2020), whose main goal is to approximate invariance to *global input transformations*. However, Augerino requires passing several augmented versions of an input during inference, and only works with continuous transformations. Partial G-CNNs on the other hand, are also capable of defining distributions on discrete sets of transformations.

To the best of our knowledge, our work is the first approach to explore partial group equivariances, and to provide a practical implementation to learn them from data end-to-end.

**Group equivariant neural networks.** The seminal work of Cohen & Welling (2016) has inspired several methods equivariant to many different groups. Existing convolutional methods show equivariance to planar rotations (Dieleman et al., 2016; Worrall et al., 2017; Weiler et al., 2018b), spherical rotations (Weiler et al., 2018a; Cohen et al., 2018b; Esteves et al., 2019a;b; 2020), scaling (Worrall & Welling, 2019; Sosnovik et al., 2019; Romero et al., 2020b), and more general symmetry groups (Romero et al., 2020a; Bogatskiy et al., 2020; Finzi et al., 2021). Group equivariant self-attention has also been proposed (Fuchs et al., 2020; Romero & Cordonnier, 2020; Hutchinson et al., 2021).

Common to all these approaches is that they require fixing the symmetry group to which the model will be *fully* equivariant prior to training. In contrast, our method allows us to define layers approximately equivariant to subsets of the group, and to learn these subsets during training.

## 3  PRELIMINARIES: GROUP EQUIVARIANCE AND THE GROUP CONVOLUTION

**Group equivariance:** Group equivariance is the property of a map to respect the transformations in a group. We say that a map is equivariant to a group if whenever the input is transformed by elements of the group, the output of the map is equally transformed but not modified. Formally, for a group $\mathcal{G}$ with elements $g \in \mathcal{G}$ acting on a set $\mathcal{X}$, and a map $\phi : \mathcal{X} \to \mathcal{X}$, we say that $\phi$ is equivariant to $\mathcal{G}$ if:

$$\phi(g \cdot x) = g \cdot \phi(x), \quad \forall x \in \mathcal{X}, \ \forall g \in \mathcal{G}. \tag{1}$$

For example, the convolution of a signal $f : \mathbb{R} \to \mathbb{R}$ and a kernel $\psi : \mathbb{R} \to \mathbb{R}$ is translation equivariant because $\mathcal{L}_t(\psi * f) = \psi * \mathcal{L}_t f$, where $\mathcal{L}_t$ translates the function by $t$: $\mathcal{L}_t f(x) = f(x - t)$. That is, if the input is translated, its numerical descriptors via the convolution are also translated but not modified.

**The group convolution.** In order to construct neural networks equivariant to a given group $\mathcal{G}$, it is crucial to define an operation that respects the symmetries in the group. The *group convolution* generalizes the convolution to be equivariant to general symmetry groups. Formally, for any $u \in \mathcal{G}$, the group convolution of a signal $f : \mathcal{G} \to \mathbb{R}$ and a kernel $\psi : \mathcal{G} \to \mathbb{R}$ is given by:

$$h(u) = (\psi * f)(u) = \int_{\mathcal{G}} \psi(v^{-1}u) f(v) \, \mathrm{d}\mu_{\mathcal{G}}(v). \tag{2}$$

where $\mu_{\mathcal{G}}(\cdot)$ is the (invariant) Haar measure of the group. The group convolution is $\mathcal{G}$-equivariant. That is, it holds that, for all $u, v, w \in \mathcal{G}$:

$$(\psi * \mathcal{L}_w f)(u) = \mathcal{L}_w (\psi * f)(u), \text{ with } \mathcal{L}_w f(u) = f(w^{-1}u)$$

**The lifting convolution.** Regularly, the input of a neural network is not readily defined on the group of interest $\mathcal{G}$, but on a sub-domain $\mathcal{X}$, i.e., $f : \mathcal{X} \to \mathbb{R}$. As a result, in order to use group convolutions, we first require an operation that *lifts* the input signal from its domain $\mathcal{X}$ to the group $\mathcal{G}$ . This operation is called a *lifting convolution*. For any $u \in \mathcal{G}$, the lifting group convolution is defined as:

$$h(u) = (\psi *_{\text{lift}} f)(u) = \int_{\mathcal{X}} \psi(v^{-1}u) f(v) \, \mathrm{d}\mu_{\mathcal{G}}(v), \ v \in \mathcal{X}, u \in \mathcal{G}. \tag{3}$$

**Practical implementation of the group convolution.** The group convolution is defined on a continuous domain. Consequently, it cannot be computed in finite time, and thus must be approximated. Two main strategies exist to approximate group convolutions with regular group representations: (*i*) group discretization (Cohen & Welling, 2016), and (*ii*) Monte Carlo approximations (Finzi et al., 2020).

Group discretization approximates the group convolution by using a fixed discretization of the group, e.g., the rotation group as the set of rotations by 90 degrees. Unfortunately, this approximation is *only* equivariant to the transformations in the discretization, and not to the underlying continuous group. A better solution is proposed by Finzi et al. (2020). Finzi et al. (2020) propose a Monte Carlo approximation of the group convolution by sampling transformations uniformly from the entire group during each forward pass. Formally, this approximation is given by:

$$h(u_i) = (\psi \hat{*} f)(u_i) = \sum_j \psi(v_j^{-1}u_i) f(v_j) \bar{\mu}_{\mathcal{G}}(v_j), \tag{4}$$

where $\{u_i\}$, $\{v_j\}$ are the discretizations of the input and output domain of the operation, respectively.

Importantly, this approximation requires the convolutional kernel $\psi$ to be defined on the *continuous group*. As the domain cannot be enumerated, $\psi$ cannot be parameterized by independent weights. Instead, Finzi et al. (2020) parameterize convolutional kernels via small neural networks. With this parameterization, all elements $v_j^{-1}u_i$ can be mapped to a well-defined kernel value.

## 4 PARTIAL GROUP EQUIVARIANT NETWORKS

In this section, we introduce our approach. First, we define the concept of partial group equivariance, and generalize the group convolution to the partial equivariant case in a constructive manner. Next, we illustrate how group subsets can be learned with learnable probability distributions on the group, and provide concrete examples for discrete groups, continuous groups and combinations thereof. We conclude by presenting the partial group convolution and the architecture of Partial G-CNNs.

### 4.1 PARTIAL GROUP EQUIVARIANCE

Contrary to group equivariance, we say that a map $\phi$ is *partially equivariant* to $\mathcal{G}$, if the map is equivariant to transformations in a subset of the group $\mathcal{S} \subset \mathcal{G}$, but not to the whole group $\mathcal{G}$:

$$\phi(g \cdot x) = g \cdot \phi(x), \quad \forall x \in \mathcal{X}, \ \forall g \in \mathcal{S}. \tag{5}$$

Different from equivariance to a *subgroup* of $\mathcal{G}$: a subset of the group which also fulfill the group axioms, we do not restrict the subset $\mathcal{S}$ to be itself a group.

As will we show in Sec. 4.2, partial equivariance can only be obtained in an approximate manner. This is because the set $\mathcal{S}$ is not necessarily closed under group actions. Partial equivariance is related to *soft invariance* (van der Wilk et al., 2018): the property of a map to be approximately invariant. We opt for the word *partial* in the equivariant setting, to make clear that it defines a subset of the group.

### 4.2 FROM GROUP CONVOLUTIONS TO PARTIAL GROUP CONVOLUTIONS

In this section, we generalize the group convolution to describe partial equivariances. Vital to our analysis is the equivariance proof of the group convolution (Cohen et al., 2018a). In addition, we must distinguish between the domains of the input and output signals of the group convolution, i.e., the domains of $f$ and $h$ in Eq. 2. This distinction is important because they might differ for partial group convolutions. From here on, we refer to these as ***input domain*** and ***output domain***.

**Proposition 4.1.** *The group convolution is $\mathcal{G}$-equivariant in the sense that for all $u, v, w \in \mathcal{G}$:*

$$(\psi * \mathcal{L}_w f)(u) = \mathcal{L}_w(\psi * f)(u), \ with \ \mathcal{L}_w f(u) = f(w^{-1}u). \tag{6}$$

*Proof.*
$$(\psi * \mathcal{L}_w f)(u) = \int_{\mathcal{G}} \psi(v^{-1}u) f(w^{-1}v) \, \mathrm{d}\mu_{\mathcal{G}}(v) = \int_{\mathcal{G}} \psi(\bar{v}^{-1}w^{-1}u) f(\bar{v}) \, \mathrm{d}\mu_{\mathcal{G}}(\bar{v})$$
$$= (\psi * f)(w^{-1}u) = \mathcal{L}_w(\psi * f)(u).$$

From the first to the second equality, we use the change of variables $\bar{v} := w^{-1}v$. We can do this because the group convolution is a map from the group to itself, and thus if $w, v \in \mathcal{G}$, so does $w^{-1}v$. Since the Haar measure is an invariant measure on the group, we have that $\mu_{\mathcal{G}}(v) = \mu_{\mathcal{G}}(\bar{v})$, for all $v, \bar{v} \in \mathcal{G}$. □

**Going from the group $\mathcal{G}$ to a subset $\mathcal{S}$.** Crucial to the proof of Proposition 4.1 is the fact the group convolution is an operation from input signals to output signals on the group. As a result, $w^{-1}u$ is a member of the output domain for any $w \in \mathcal{G}$ applied to the input domain. Consequently, a group transformation applied to the input can be reflected by an equivalent transformation on the output.

Now, consider the case in which the output domain is not the group $\mathcal{G}$, but instead an arbitrary subset $\mathcal{S} \subset \mathcal{G}$, e.g., rotations in $[-\frac{\pi}{2}, \frac{\pi}{2}]$. Following the proof of Proposition 4.1 with $u \in \mathcal{S}$, and $v \in \mathcal{G}$, we observe that the operation is equivariant to transformations $w \in \mathcal{G}$ as long as $w^{-1}u$ is a member of $\mathcal{S}$. However, if $w^{-1}u$ does not belong to the output domain $\mathcal{S}$, the output of the operation cannot reflect an equivalent transformation to that of the input, and thus equivariance is not guaranteed.

Note, however, that equivariance is *only* obtained if Eq. 6 holds for *all* elements in the output domain. That is, if $w^{-1}u$ is a member of $\mathcal{S}$, for all elements $u \in \mathcal{S}$. This is not the case in general, as the output domain $\mathcal{S}$ is not necessarily closed under group transformations. It is this difference, in fact, what Partial G-CNNs make use of to disrupt full equivariance whenever necessary (Fig. 1).

*Quantifying the equivariance difference.* Importantly, we can precisely quantify how much the output response will change for any input transformation given an output domain $\mathcal{S}$. Intuitively, this difference is given by the difference in the parts of the output feature representation that go in and out of $\mathcal{S}$ by the action of the input transformation. Naturally, the larger the input transformation and the smaller the size of $\mathcal{S}$, the larger the equivariance difference will be. Fig. 1 illustrates this effect. A formal discussion and the derivation of the equivariance difference are provided in Appx. B.1.

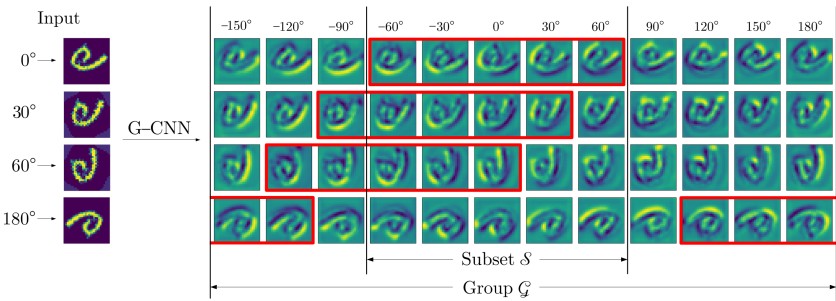

Figure 1: Partial group convolution. In a group convolution, the domain of the output is closed under group transformations of the input. Consequently, all response components are part of the output regardless of the transformation applied to the input. Differently, the output domain $\mathcal{S}$ of a partial group convolution is not necessarily closed under input transformations. As a result, the feature representation captured within $\mathcal{S}$ will change for different transformations of the input. In Fig. 1, for instance, the output feature representation of the input (outlined in red) gradually leaves $\mathcal{S}$ for different transformations of the input. For large transformations (180° here), the responses within $\mathcal{S}$ change entirely. This difference allows partial group convolutions to distinguish among input transformations.

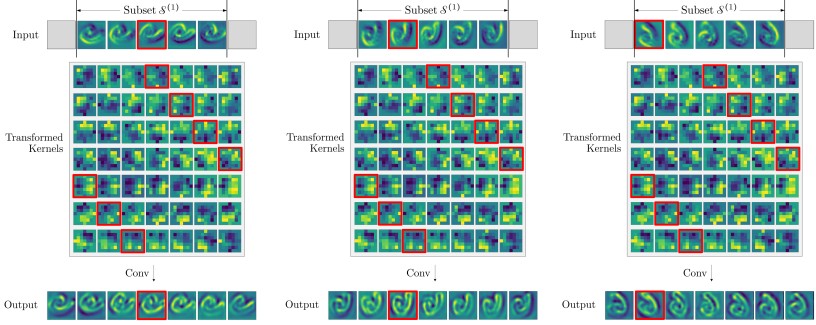

Figure 2: Partial group convolution on a group subset. Contrary to the group convolution, partial group convolutions can receive an input whose domain is not the entire group $\mathcal{G}$, but a subset thereof $\mathcal{S}^{(1)}$. Consequently, the input feature representation falling within $\mathcal{S}^{(1)}$ may change for different input transformations (top). As a result, the output of the partial group convolution may show variations for different input transformations even if the output domain of the group convolution is the group $\mathcal{G}$ (bottom row). Similar to the case in Fig. 1, this difference grows proportionally to the strength of the input transformation. This can be seen in the red-outlined feature representations at the bottom.

**Going from a subset $\mathcal{S}^{(1)}$ to another subset $\mathcal{S}^{(2)}$.** Now, consider the case in which the domain of the input and the output of the group convolution are both subsets of the group, i.e., $v \in \mathcal{S}^{(1)}$ and $u \in \mathcal{S}^{(2)}$. Analogous to the previous case, equivariance to input transformations $w \in \mathcal{G}$ holds at positions $u \in \mathcal{S}^{(2)}$ for which $w^{-1}u$ are also members of $\mathcal{S}^{(2)}$. Now however, the input domain is not restricted to be closed. Consequently, the feature representation of the input will differ for different group transformations. This induces an additional difference in the output response.

To see this, consider the partial group convolution in Fig. 2. For different input transformations, the feature representations within $\mathcal{S}^{(1)}$ change. As a result, the output feature representation might change for different input transformations, even if the output domain is the group itself, i.e., $\mathcal{S}^{(2)}{=}\mathcal{G}$.

In summary, considering a group convolution from a subset $\mathcal{S}^{(1)}$ to a subset $\mathcal{S}^{(2)}$ has two sources of equivariance difference. The first, comes from considering a subset $\mathcal{S}^{(2)}$ as the output domain, and is equivalent to the difference illustrated in the previous case. The second, comes from considering a subset $\mathcal{S}^{(1)}$ in the input domain. This allows the input features to change as a function of the input transformation, and the size of the input domain $\mathcal{S}^{(1)}$. This in turn induces an additional difference in the output response. A formal treatment of the second difference term is provided Appx. B.2.

**Going from a subset $\mathcal{S}$ to the group $\mathcal{G}$.** Finally, we can also consider the case in which the input domain is a subset $\mathcal{S}$ of the group and the output domain is the group itself $\mathcal{G}$. Following the previous analysis, we can see that, this operation is equivariant to all input transformations $w \in \mathcal{G}$, as $w^{-1}u$ is

always member of the output domain for all values of $u \in \mathcal{G}$. Nevertheless, the input features can change for different input transformations, and thus the difference illustrated in the previous case can also emerge here. This case can be seen as an special case of the previous section, where $\mathcal{S}^{(2)} = \mathcal{G}$, and thus the equivariance difference induced by considering a subset in the output domain is zero.

Interestingly, the lifting convolution is an special case of this instance with $\mathcal{S} = \mathcal{X}$. Note however, that the lifting convolution is exactly equivariant. This is because the input domain of the operation is $\mathcal{X}$. Consequently, the input domain remains equal for all input transformations.

## 4.3 Learning Group Subsets via Probability Distributions on Group Elements

So far, we have discussed the properties of the partial group convolution without providing details on how to learn group subsets during training. In this section, we describe our approach and provide specific examples for discrete groups, continuous groups, and combinations thereof.

Vital to our approach is the Monte Carlo approximation to the group convolution presented in Sec. 3:

$$(\psi \,\hat{\star}\, f)(u_i) = \sum_j \psi(v_j^{-1} u_i) f(v_j) \bar{\mu}_{\mathcal{G}}(v_j).$$

As shown in Appx. C, this approximation is equivariant to $\mathcal{G}$ in expectation if the elements in the input and output domain, $\{u_i\}, \{v_j\}$ are uniformly sampled from the Haar measure, i.e., $u_i, v_j \sim \mu_{\mathcal{G}}(\cdot)$.[1]

**Approach.** Our main observation is that we can prioritize sampling specific group elements during the group convolution by learning a probability distribution $\mathrm{p}(g)$ over the elements of the group.

When group convolutions draw elements uniformly from the entire group, each group element is drawn with equal probability and thus, the resulting approximation is fully equivariant in expectation. However, we can also define a different probability distribution with which some samples are drawn with larger probability. For instance, we can sample from a predefined region, e.g., rotations in $\left[-\frac{\pi}{2}, \frac{\pi}{2}\right]$, by defining a probability distribution $\mathrm{p}(g)$ which is uniform in this range, but zero otherwise. In the limit, we can forget a certain equivariance by letting this distribution collapse to a single point, e.g., the identity, along the corresponding group dimension.

In other words, learning a proper probability distribution $\mathrm{p}(g)$ on the group can be used to effectively learn a subset of the group. Based on this intuition, we define a probability distribution $\mathrm{p}(g)$ on the output domain of the operation, i.e., on $\{u_i\}$, in order to learn a subset of the group $\mathcal{S}^{(2)}$ upon which partial equivariance is defined. Note that we only need to learn a distribution on the output domain for each layer. This is because group convolutional layers are applied sequentially, and thus the subset of the input domain is learned by a distribution on the output domain of the previous layer.

### 4.3.1 Defining $\mathrm{p}(g)$ for Discrete, Continuous and Multi-Dimensional Groups

**Probability distributions for one-dimensional continuous groups.** We take inspiration from Augerino (Benton et al., 2020), an use the reparameterization trick (Kingma & Welling, 2013) to parameterize continuous distributions. In particular, we use the reparameterization trick on the Lie algebra of the group (Falorsi et al., 2019) to define a distribution which is uniform over a connected set of group elements $[u^{-1}, \ldots, e, \ldots, u]$, but zero otherwise. To this end, we define a distribution $\mathfrak{u} \cdot [-1, 1]$ with learnable $\mathfrak{u}$ on the Lie algebra $\mathfrak{g}$, and map it to the group via the pushforward of the exponential map $\exp : \mathfrak{g} \to \mathcal{G}$. This give us a distribution which is uniform over a connected set of elements $[u^{-1}, \ldots, e, \ldots, u]$, but zero otherwise.[2]

For instance, we can learn a distribution on the rotation group $\mathrm{SO}(2)$, which is uniform between $[-\theta, \theta]$ and zero otherwise by defining a probability distribution $\theta \cdot [-1, 1]$ with learnable $\theta$ on the Lie algebra, and mapping it to the group. If we parameterize group elements as scalars $g \in [-\pi, \pi)$, the exponential map is the identity, and thus $\mathrm{p}(g) = \mathcal{U}(\theta \cdot [-1, 1))$. If we sample group elements from this distribution during the calculation of the group convolution, the output domain will only contain elements in $[-\theta, \theta)$ and the output feature map will be partially equivariant.

**Probability distributions for one-dimensional discrete groups.** We can define a probability distribution on a discrete group as the probability of sampling from all possible element combinations.

---

[1] Finzi et al. (2020) show a similar result where $u_i$ and $v_j$ are the same points and thus $v_j \sim \mu_{\mathcal{G}}(\cdot)$ suffices.
[2] Note that an $\exp$-pushforwarded local uniform distribution is locally equivalent to the Haar measure, and thus we can still use the Haar measurement for integration on group subsets.

For instance, for the mirroring group $\{1, -1\}$, this distribution assigns a probability to each of the combinations $\{0,0\}, \{0,1\}, \{1,0\}, \{1,1\}$ indicating whether the corresponding element is sampled or not. For a group with elements $\{e, g_1, \ldots, g_n\}$, however, this oblige us to sample from $2^{n+1}$ elements, which is computationally expensive and potentially unstable to train. Instead, we define element-wise Bernoulli distributions over the elements $\{g_1, \ldots, g_n\}$, and learn the probabilities $\{p_i\}$ of sampling the elements $\{g_i\}$. The probability distribution on the group can be then formulated as the joint probability of the element-wise Bernoulli distributions $\mathrm{p}(e, g_1, \ldots, g_n) = \prod_{i=1}^{n} \mathrm{p}(g_i)$.

To learn the element-wise distributions, we make use of the Gumbel-Softmax trick (Jang et al., 2016; Maddison et al., 2016), and use the Straight-Through Gumbel-Softmax to back-propagate through sampling. If all the probabilities $\{p_i{=}1\}$, the group convolution will be fully equivariant. However, whenever some probabilities start declining, group equivariance becomes partial. In the limit, if all probabilities $\{p_i{=}0\}$, the operation effectively forgets this equivariance.

**Probability distributions for multi-dimensional groups.** Multi-dimensional groups are important for the applications we consider. For instance, the orthogonal group $\mathrm{O}(2)$ is parameterized by rotations and mirroring, or the dilation-rotation group by scaling and rotations. In such cases, we construct the probability distribution over the entire group as a combination of *independent probability distributions along each of these axes*. For a group $\mathcal{G}$ with elements $g$ decomposable along $n$ dimensions $g{=}(d_1, ..., d_n)$, we decompose the probability distribution as: $\mathrm{p}(g){=}\prod_{i=1}^{n} \mathrm{p}(d_i)$, where the probability $\mathrm{p}(d_i)$ is defined given the type of space, i.e., continuous or discrete. For instance, for the orthogonal group $\mathrm{O}(2)$ defined by rotations $r$ and mirroring $m$, i.e., $g = (r, m)$, $r \in \mathrm{SO}(2), m \in \{\pm 1\}$, we define the probability distribution on the group as $\mathrm{p}(g){=}\mathrm{p}(r){\cdot}\mathrm{p}(m)$, where $\mathrm{p}(r)$ is a continuous distribution, and $\mathrm{p}(m)$ is a discrete one.

### 4.4 THE PARTIAL GROUP CONVOLUTION

Let $\mathcal{S}^{(1)}, \mathcal{S}^{(2)}$ be subsets of a group $\mathcal{G}$ and $\mathrm{p}(u)$ be uniform in $\mathcal{S}^{(2)}$ and 0 otherwise. The partial group convolution from a function $f : \mathcal{S}^{(1)} \to \mathbb{R}$ to a function $h : \mathcal{S}^{(2)} \to \mathbb{R}$ is given by:

$$h(u) = (\psi \star f)(u) = \int_{\mathcal{S}^{(1)}} \mathrm{p}(u)\psi(v^{-1}u)f(v)\,\mathrm{d}\mu_{\mathcal{G}}(v); \ u \in \mathcal{S}^{(2)}, v \in \mathcal{S}^{(1)}. \tag{7}$$

The cases described in Sec. 4.2 can be written as special cases of Eq. 7. In practice, Eq. 7 is computed with a Monte Carlo approximation as described in Algorithm 1 (Appx. D).

### 4.5 PARTIAL GROUP EQUIVARIANT NETWORKS

We built upon the work of Finzi et al. (2020) and construct base G-CNNs equivariant to continuous and discrete groups. This is achieved by combining kernel parameterizations on Lie groups, and actions of discrete groups on the group representation of the kernels. Moreover, we replace the isotropic lifting layer of Finzi et al. (2020) by a convolutional layer with lifting convolutions (Eq. 3). Inspired by Romero et al. (2021), we parameterize convolutional kernels as implicit neural representations with SIRENs (Sitzmann et al., 2020). The proposed parameterization shows better expressivity and convergence speed, and outperforms ReLU-, LeakyReLU and Swish-MLP parameterizations used so far for continuous G-CNNs, e.g., Schütt et al. (2017); Finzi et al. (2020), Tab. 4. Our network structure is shown in Fig. 3, and varies only in the number of blocks and channels.

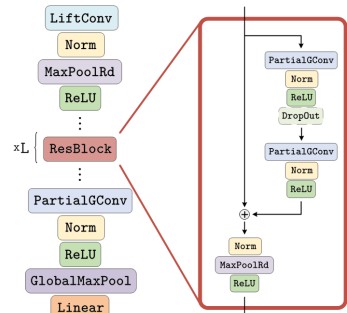

Figure 3: Partial G-CNN architecture.

## 5 EXPERIMENTS

**Experimental details.** We parameterize all our convolutional kernels as a 3-layer SIREN (Sitzmann et al., 2020) with 32 hidden units. All our networks are constructed with two residual blocks of 32 channels each, batch normalization (Ioffe & Szegedy, 2015) and follow the structure shown in Fig. 3. Our networks are intentionally selected to be simple to better assess the effect of partial equivariance. Importantly, we avoid learning probability distributions along the translation group, and assume all spatial positions to be sampled. This allows us to use fast `PyTorch` primitives in our implementation.

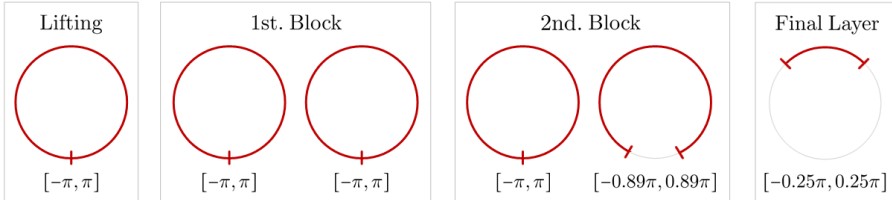

Figure 4: Learned full/partial equivariances for MNIST6-180 with a Partial SE(2)-ResNet.

Table 1: Test accuracy on MNIST6-180 and MNIST6-M.

| MODEL | BASE GROUP | MNIST6-180 ACC. (%) | BASE GROUP | MNIST6-M ACC. (%) | BASE GROUP | MNIST6-M ACC. (%) |
|---|---|---|---|---|---|---|
| G-CNN | SE(2) | 50.0 | Mirroring | 50.0 | E(2) | 50.0 |
| Partial G-CNN | | **100.0** | | **100.0** | | **100.0** |

**Toy tasks: MNIST6-180 and MNIST6-M.** First, we validate the ability of our approach to solve situations in which full equivariance impedes solving a task. To this end, we construct two datasets: *MNIST6-180*, and *MNIST6-M*, by extracting 6 numbers from the MNIST dataset (LeCun et al., 1998), and modifying them either by 180° rotations, or mirroring. The original numbers get the label "original", and the transformed data the label "transformed". The goal is to separate these two classes.

G-CNNs are unable to solve tasks for which discrimination among group transformations are required. As a result, SE(2)-CNNs are unable to separate MNIST6-180 classes, and E(2)-CNNs are unable to separate MNIST6-M classes (Tab 1). Partial G-CNNs with base groups SE(2), E(2), on the other hand, seamlessly solve the tasks. Interestingly, we observe that the original MNIST6 class already contains some mirrored numbers, e.g., the input used in Fig. 1, and thus some class overlap exists in MNIST6-M. Nevertheless, Partial G-CNNs are able to solve the task to perfection.

We can also visualize the learned full/partial equivariances at every layer of a network. Fig. 4 illustrates the learned equivariances for a Partial SE(2)-CNN with 2 residual blocks trained on MNIST6-180. In order to solve the task, the network learns to progressively reduce its equivariance as a function of depth, up to a final range of $[-0.25\pi, 0.25\pi]$. Interestingly however, the network does not disrupt equivariance directly at the beginning of the network, but instead decides to stay fully equivariant in the first four layers. We hypothesize that this is because these layers extract low- and mid-level features, which might appear at arbitrary poses. As a consequence, it is advantageous to preserve full equivariance at this hierarchical level. The same behaviour is also present in E(2)-CNNs. However, we observe that discrete distributions are more unstable to train, and other patterns also emerge that, for example, interrupt full equivariance at the second or third layer of the network.

**Benchmark image datasets.** Next, we validate our approach on the following benchmark image recognition datasets: rotated MNIST (Larochelle et al., 2007), CIFAR-10 and CIFAR-100 (Krizhevsky et al., 2009). Additional results on PatchCamelyon (Veeling et al., 2018) can be found in Appx. F. We construct Partial G-CNNs with base groups SE(2) and E(2), and varying number of elements used in the Monte Carlo approximation, and compare them to equivalent ResNets and G-CNNs.

Our results are shown in Table 2. Partial G-CNNs are competitive with G-CNNs where full-equivariance is needed, i.e., RotMNIST and PatchCamelyon. However, for tasks in which the data does not naturally exhibit full rotation equivariance, i.e., CIFAR-10 and CIFAR-100, Partial G-CNNs consistently outperform their full equivariant counterparts. Our results support the usage of Partial G-CNNs for settings with both partial and full equivariance.

In addition, we validate our claim that SIRENs are better suited to parameterize group convolutional kernels than existing alternatives. As shown in Tab. 4, SE(2)-CNNs with SIREN kernels consistently outperform SE(2)-CNNs with kernel parameterizations via ReLU, LeakyReLU and Swish nonlinearities on all the image benchmarks considered. Our results support the usage of SIRENs as parameterization for continuous group convolutional kernels.

## 6 DISCUSSION

**Memory consumption in partial equivariant settings.** G-CNNs fix the number of samples with which the group convolution is approximated prior to training. G-CNNs use the same number of samples for all group convolutional layers regardless of the symmetries present in the data. Partial

Table 2: Test accuracy on vision benchmark datasets.

| Model | Base Group | No. Elements | Partial Equiv. | Classification Accuracy (%) | | |
|---|---|---|---|---|---|---|
| | | | | RotMNIST | CIFAR-10 | CIFAR-100 |
| ResNet | T(2) | 1 | - | 97.23 | 83.11 | 47.99 |
| G-CNN | SE(2) | 4 | ✗ | 99.10 | 83.73 | 52.35 |
| | | | ✓ | **99.13** | **86.15** | **53.91** |
| | | 8 | ✗ | 99.17 | 86.08 | 55.55 |
| | | | ✓ | **99.23** | **88.59** | **57.26** |
| | | 16 | ✗ | **99.24** | 86.59 | 51.55 |
| | | | ✓ | 99.18 | **87.45** | **57.31** |
| | E(2) | 8 | ✗ | **98.14** | 85.55 | 54.29 |
| | | | ✓ | 97.78 | **89.00** | **55.22** |
| | | 16 | ✗ | 98.35 | 88.95 | 57.78 |
| | | | ✓ | **98.58** | **90.12** | **61.46** |

G-CNNs on the other hand, are able to *dynamically* adjust the number of samples used in order to approximate partial group convolutions. In particular, we fix a maximum number of samples prior to training, and restrict the amount of samples used at every layer based on the subset of the group learned. For instance, a Partial SE(2)-CNN with a learned distribution $p(u)=\mathcal{U}(\frac{\pi}{2}[-1,1])$ only uses half of the elements used in the corresponding G-CNN. This efficiency in memory leads to important reductions in training time for Partial G-CNNs on partial equivariant settings.

**Better kernel parameterization.** Inspired by Romero et al. (2021), we replace the Swish kernel parameterization proposed by Finzi et al. (2020) by a SIREN Sitzmann et al. (2020). We observe that this parameterization consistently leads to better results and faster convergence.

**Sampling per batch element.** In our experiments, we sample once from the learned distribution $p(u)$ at every layer, and use this sample for all elements in the batch. A better estimation of the probability distribution could be obtained by drawing a different sample for each element in the batch at every layer. Although this method may benefit the learned distributions, it comes at a large memory cost. Consequently, all our reported results are obtained using our original formulation.

**Partial equivariances for other group representations.** Our theory of learnable partial equivariances is directly applicable to methods using regular group representations, e.g., Cohen & Welling (2016); Romero & Cordonnier (2020); Hutchinson et al. (2021). An extension for other kind of group representations such as irreducible representations in steerable CNNs, e.g., (Worrall et al., 2017; Weiler et al., 2018b; Weiler & Cesa, 2019) is an interesting research direction, as irreducible representations allow for exact equivariance to continuous groups.

**Unstable training on discrete groups.** Although discrete groups can be successfully modelled in partial equvariance settings, we observe that the discrete distribution used here is not very stable to train. Finding good parameterizations to learn discrete distributions is an active field of research (Hoogeboom et al., 2019; 2021), and advances in this field could lead to more stable training of partial equivariances on discrete groups.

## 7 Conclusion and Future Work

We presented Partial Group equivariant Convolutional Neural Networks (Partial G-CNNs): a new family of equivariant methods able to learn partial and full equivariances directly from data at every layer. Partial G-CNNs match the performance of G-CNNs in full equivariant settings, and outperform G-CNNs in settings in which partial group equivariance better represents the data. By learning different levels of equivariance at every layer, Partial G-CNNs are able to maintain full group equivariance in early layers to efficiently learn low-level features appearing at arbitrary poses. However, they are able to restrict it whenever necessary to discriminate among high-level features.

Our method entirely relies on a good parameterization of the probability distribution defined over group elements. Finding better representations and learning methods for these distributions, specially for distributions on discrete variables, is a promising direction for future research. In addition, an equally interesting future research direction is the extension of partial equivariances to neural architectures handling other kind of group representations such as steerable CNNs.

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

# APPENDIX

## A   GROUPS AND GROUP ACTION

**Groups:** Group theory is the mathematical language that describes symmetries. The core mathematical object is that of a *group*, and defines what it means for something to exhibit symmetries. Specifically, a group consists of a tuple $(\mathcal{G}, \cdot)$ – set of transformations $\mathcal{G}$, and a binary operation $\cdot$ with the following properties: (i) closure, i.e., $g_1 \times g_2 = g_3 \in \mathcal{G}, \forall g_1, g_2 \in \mathcal{G}$ (ii) associativity, i.e., $g_1 \cdot (g_2 \cdot g_3) = (g_1 \cdot g_2) \cdot g_3$ for all $g_1, g_2, g_3 \in \mathcal{G}$, (iii) the existence of an identity element $e \in \mathcal{G}$, such that $g \cdot e = e \cdot g = g$, and (iv) the existence of an inverse $g^{-1} \in \mathcal{G}$ for all $g \in \mathcal{G}$.

**Group action:** One can define the *action of the group* $\mathcal{G}$ on a set $\mathcal{X}$. This action describes how group elements $g \in \mathcal{G}$ modify the set $\mathcal{X}$ when the transformation is applied. For instance, the action of elements in the group of planar rotations $\theta \in \mathrm{SO}(2)$ on an image $x \in \mathcal{X}$ –written $\theta x$–, depicts how the image $x$ changes when the rotation $\theta$ is applied.

## B   FORMAL TREATMENT OF THE EQUIVARIANCE DIFFERENCE IN PARTIAL GROUP CONVOLUTIONS

### B.1   PARTIAL GROUP CONVOLUTIONS FROM THE GROUP $\mathcal{G}$ TO A SUBSET $\mathcal{S}$

The partial group convolution from signals on $\mathcal{G}$ to signals on a subset $\mathcal{S}$ can be interpreted as a group convolution for which the output signal outside of $\mathcal{S}$ is set to zero. Consequently, we can calculate the equivariance difference $\Delta_{\mathrm{equiv}}$ in the feature representation, by calculating the difference on the subset $\mathcal{S}$ of a group convolution with a group-transformed input $(\mathcal{L}_w f * \psi)$ and a group convolution with a canonical input proceeded by the same transformation on $\mathcal{S}$, i.e., $\mathcal{L}_w(f * \psi)$.

The equivariance difference given by the effect of a subset in the output domain $\Delta_{\mathrm{equiv}}^{\mathrm{out}}$ is given by:

$$
\begin{aligned}
\Delta_{\mathrm{equiv}}^{\mathrm{out}} &= \left\| \int_{\mathcal{S}} \mathcal{L}_w(\psi * f)(u)\, \mathrm{d}\mu_{\mathcal{G}}(u) - \int_{\mathcal{S}} (\psi * \mathcal{L}_w f)(u)\, \mathrm{d}\mu_{\mathcal{G}}(u) \right\|_2^2 \\
&= \left\| \int_{w^{-1}\mathcal{S}} (\psi * f)(w^{-1}u)\, \mathrm{d}\mu_{\mathcal{G}}(u) - \int_{\mathcal{S}} (\psi * f)(w^{-1}u)\, \mathrm{d}\mu_{\mathcal{G}}(u) \right\|_2^2 \\
&= \left\| \int_{\mathcal{S}} (\psi * f)(u)\, \mathrm{d}\mu_{\mathcal{G}}(u) - \int_{w\mathcal{S}} (\psi * f)(u)\, \mathrm{d}\mu_{\mathcal{G}}(u) \right\|_2^2 \\
&= \left\| \int_{s_{\min}}^{s_{\max}} (\psi * f)(u)\, \mathrm{d}\mu_{\mathcal{G}}(u) - \int_{ws_{\min}}^{ws_{\max}} (\psi * f)(u)\, \mathrm{d}\mu_{\mathcal{G}}(u) \right\|_2^2 \\
&= \left\| \left( \int_{ws_{\max}}^{s_{\max}} (\psi * f)(u)\, \mathrm{d}\mu_{\mathcal{G}}(u) + \int_{s_{\min}}^{ws_{\max}} (\psi * f)(u)\, \mathrm{d}\mu_{\mathcal{G}}(u) \right) - \right. \\
&\qquad \left. \left( \int_{s_{\min}}^{ws_{\max}} (\psi * f)(u)\, \mathrm{d}\mu_{\mathcal{G}}(u) + \int_{ws_{\min}}^{s_{\min}} (\psi * f)(u)\, \mathrm{d}\mu_{\mathcal{G}}(u) \right) \right\|_2^2 \\
&= \left\| \int_{ws_{\max}}^{s_{\max}} (\psi * f)(u)\, \mathrm{d}\mu_{\mathcal{G}}(u) - \int_{ws_{\min}}^{s_{\min}} (\psi * f)(u)\, \mathrm{d}\mu_{\mathcal{G}}(u) \right\|_2^2
\end{aligned}
$$

From the first line to the second we take advantage of the equivariance property of the group convolution: $(\mathcal{L}_w f * \psi)(u) = \mathcal{L}_w(f * \psi)(u)$, and account for the fact that only the region within $\mathcal{S}$ is visible at the output. We use the change of variables $u = w^{-1}u$ from the second to third line. We specify the boundaries of $\mathcal{S}$ from the third to the fourth line. In the fifth line we separate the integration over $\mathcal{S}$ as a sum of two integrals which depict the same range. In the last line, we cancel out the overlapping parts of the two integrals to come to the final result.

In conclusion, the equivariance difference induced by a subset $\mathcal{S}^{(2)}$ on the domain of the output $\Delta_{\mathrm{equiv}}^{\mathrm{out}}$ is given by the difference between the part of the representation that leaves the subset $\mathcal{S}$, and the part that comes to replace it instead. This behaviour is illustrated in Figure 1.

### B.2 PARTIAL GROUP CONVOLUTIONS FROM A SUBSET $\mathcal{S}^{(1)}$ TO A SUBSET $\mathcal{S}^{(2)}$

To isolate the effect of having a group subset as domain of the input signal $f$, we first consider the domain of the output to be the group, i.e., $\mathcal{S}^{(2)}=\mathcal{G}$. The equivariance difference is given by the difference across the entire output representation of the group convolution calculated on an input subset $\mathcal{S}^{(1)}$ with a canonical input $f$, and with a group transformed input $\mathcal{L}_w f$.

The equivariance difference $\Delta_{\text{equiv}}^{\text{in}}$ given by the effect of a subset in the input domain is given by:

$$\Delta_{\text{equiv}} = \left\| \int_{\mathcal{G}} \int_{\mathcal{S}} \psi(v^{-1}u)f(v)\,\mathrm{d}\mu_{\mathcal{G}}(v)\mathrm{d}\mu_{\mathcal{G}}(u) - \int_{\mathcal{G}} \int_{\mathcal{S}} \psi(v^{-1}u)f(w^{-1}v)\,\mathrm{d}\mu_{\mathcal{G}}(v)\mathrm{d}\mu_{\mathcal{G}}(u) \right\|_2^2$$

$$= \left\| \int_{\mathcal{G}} \left[ \int_{\mathcal{S}} \psi(v^{-1}u)f(v)\,\mathrm{d}\mu_{\mathcal{G}}(v) - \int_{\mathcal{S}} \psi(v^{-1}u)f(w^{-1}v)\,\mathrm{d}\mu_{\mathcal{G}}(v) \right] \mathrm{d}\mu_{\mathcal{G}}(u) \right\|_2^2$$

$$= \left\| \int_{\mathcal{G}} \int_{\mathcal{S}} \psi(v^{-1}u) \left[ f(v) - f(w^{-1}v) \right] \mathrm{d}\mu_{\mathcal{G}}(v)\mathrm{d}\mu_{\mathcal{G}}(u) \right\|_2^2$$

In other words, the equivariance difference induced by a subset $\mathcal{S}^{(1)}$ on the domain of the input $\Delta_{\text{equiv}}^{\text{in}}$ is given by the difference in $\mathcal{S}^{(1)}$ of the canonical input $f$, and the part that comes to replace it when the input is modified by a group transformation $w$.

## C    EQUIVARIANCE PROPERTY OF MONTE-CARLO APPROXIMATIONS

Consider the Monte-Carlo approximation depicted in Eq. 4:

$$(\psi \hat{\star} f)(u_i) = \sum_j \psi(v_j^{-1}u_i)f(v_j)\bar{\mu}_{\mathcal{G}}(v_j).$$

For a transformed version of the $\mathcal{L}_w f$, we can show that the Monte-Carlo approximation of the group convolution is equivariant in expectation. The proof follows the same steps than Finzi et al. (2020). However, the last step of the proof has a different reason, resulting from the fact that input and output elements can be sampled from different distributions.

For a transformed version of the $\mathcal{L}_w f$, we have that:

$$(\psi \hat{\star} \mathcal{L}_w f)(u_i) = \sum_j \psi(v_j^{-1}u_i)f(w^{-1}v_j)\bar{\mu}_{\mathcal{G}}(v_j)$$

$$= \sum_j \psi(\tilde{v}_j^{-1}w^{-1}u_i)f(\tilde{v}_j)\bar{\mu}_{\mathcal{G}}(\tilde{v}_j)$$

$$\overset{d}{=} (\psi \hat{\star} f)(w^{-1}u_i) = \mathcal{L}_w(\psi \hat{\star} f)(u_i)$$

From the first to the second line, we use the change of variables $\tilde{v}_j = wv_j$ and the fact that, group elements in the input domain are sampled from the Haar measure for which it holds that $\bar{\mu}_{\mathcal{G}}(v_j)=\bar{\mu}_{\mathcal{G}}(\tilde{v}_j)$. However, from the second to the third line, we must also assume that this holds for the output domain. That is, that the probability of drawing $w^{-1}u_j$ is equal to that of drawing $u_j$. This is of particular importance for the partial equivariance setting, in which this might not be the case.

## D    ALGORITHM FOR MONTE-CARLO APPROXIMATION OF THE PARTIAL GROUP CONVOLUTION

---

**Algorithm 1:** The Partial Group Convolution Layer

---

**Inputs:** position, function-value tuples on the group or a subset thereof $\{v_j, f(v_j)\}$.
**Outputs:** convolved position, function-value tuples on the output group subset $\{u_i, (f \hat{\star} \psi)(u_i)\}$.
$\{u_i\} \sim \mathrm{p}(u)$; /* Sample elements from p(u)
**for** $u_i \in \{u_i\}$ **do**
 $h(u_i) = \sum_j \psi(v_j^{-1}u_i)f(v_j)\bar{\mu}_{\mathcal{G}}(v_j)$; /* Compute group convolution (Eq. 4)
**end**
**return** $\{u_i, h(u_i)\}$

---

Table 3: Image recognition accuracy on PatchCam dataset.

| Model | Base Group | No. Elements | Partial Equiv. | Classification Accuracy on PatchCam (%) |
|---|---|---|---|---|
| ResNet | T(2) | 1 | - | 67.59 |
| G-CNN | SE(2) | 8 | ✗ | **89.87** |
| | | | ✓ | 89.07 |
| | | 16 | ✗ | 89.71 |
| | | | ✓ | **90.31** |
| | E(2) | 16 | ✗ | **89.77** |
| | | | ✓ | 88.13 |

## E  ARCHITECTURE AND HYPERPARAMETER DETAILS

We note that all the hyperparameters were chosen based on the best performance for the fully-equivariant networks and the same parameters were used for training the proposed Partial G-CNNs.

For all the experiments shown in this paper, we use residual networks (shown in the main paper in Figure 3) with an initial lifting convolutional layer followed by 2 ResBlocks with full/partial group convolutional layers (or regular convolutional layers for the non-equivariant ResNet baselines). For all the networks for all the datasets, we use 32 feature maps in the hidden layers. We employ Batch Normalization and ReLU non-linearities as shown in the figure. For MNIST6-M and MNIST6-180, max-pooling is performed after each of the ResBlock partial/full group convolutional layers. In the case of rotMNIST, max-pooling is performed after the lifting convolutional layer and the first group convolutional layers. For CIFAR-10 and CIFAR-100, we use max-pooling layers after each of the group convolutional layers. Finally, for PatchCamelyon, we apply the max-pooling operation after the lifting convolution as well as the both the group convolutional layers. At the end of the network, a global max-pooling layer is used to create the invariant before classification.

The networks for MNIST6-180, MNIST6-M, rotMNIST, CIFAR-10 and CIFAR-100 are trained for 300 epochs and the networks for PatchCamelyon are trained for 30 epochs. We use a batch size of 64 for all networks. We use an initial learning rate of $1e-3$ and Adam optimizer with cosine annealing with 5 epochs of linear warm-up. In the case of CIFAR-10, CIFAR-100 and PatchCamelyon datasets, we also use a weight decay of $0.0001$. For roMNIST, CIFAR-10, CIFAR-100 and PatchCamelyon, we also use a different learning rate of $1e-4$ for only the parameters of the probability distributions over the group elements.

## F  ADDITIONAL EXPERIMENTAL RESULTS

**Classification results on PatchCamelyon.** Table 3 shows the results obtained for ResNet, G-CNNs and Partial G-CNNs on the PatchCamelyon dataset (Veeling et al., 2018). Partial G-CNNs match the performance of G-CNNs in this full equivariant setting, as expected.

We observe that the learned distributions over the group elements for rotMNIST and PatchCamelyon are exactly consistent with full-equivariance, and for CIFAR-10 and CIFAR-100, they clearly show partial equivariance

**Evaluation of existing kernel parameterizations.** We observe that SIREN kernels consistently lead to better accuracy than other existing kernel parameterizations. Our results are shown in Tab. 4.

Table 4: Comparison of kernel parameterizations.

| Model | No. Elements | Kernel Type | Classification Accuracy (%) | | |
|---|---|---|---|---|---|
| | | | RotMNIST | CIFAR-10 | CIFAR-100 |
| SE(2)-CNN | 4 | ReLU | 96.49 | 59.95 | 28.01 |
| | | LeakyReLU | 94.47 | 56.19 | 27.36 |
| | | Swish | 94.41 | 66.12 | 34.20 |
| | | SIREN | **99.10** | **83.73** | **52.35** |
| | 8 | ReLU | 97.73 | 68.29 | 37.81 |
| | | LeakyReLU | 97.65 | 68.94 | 36.30 |
| | | Swish | 97.72 | 69.20 | 34.10 |
| | | SIREN | **99.17** | **86.08** | **55.55** |
| | 16 | ReLU | 98.49 | 66.84 | 37.72 |
| | | LeakyReLU | 98.53 | 68.01 | 38.29 |
| | | Swish | 98.55 | 65.99 | 37.72 |
| | | SIREN | **99.24** | **86.68** | **51.51** |

