# OpenReview forum: "Learning Equivariances and Partial Equivariances From Data"
_ICLR.cc/2022/Conference — ICLR 2022 Submitted_

### Official Review · Reviewer_xU2c · 2021-10-28

**Correctness:** 2
**Technical Novelty And Significance:** 2
**Empirical Novelty And Significance:** 2
**Recommendation:** 5
**Confidence:** 5

**Main Review:**

**Strengths**
The main strength of the paper is that idea is conceptually simple and well executed. Indeed, enforcing strict equivariances is often a strong constraint in learning problems and as a result the ability to toggle the amount of equivariance needed should prove beneficial. The writing in this paper is also well done for the most part and the overall presentation is of high quality.

**Weaknesses**
In my opinion there are a few weaknesses in this paper that I believe the authors could address.

***Presentation Issues:***
Starting with presentation, I found the notation to be unclear or inconsistent at times. For example, in Eqn 3. $f$ as previously defined is a map from $G\to \mathbb{R}$ but $v \in X$. The lifting operation as a result is bit confusing. I'm aware of the lifting operation as defined in Kondor & Trivedi 2018 [1], can you please comment whether this is the same thing being done here? Also, Eqn 3. assumes a continuous/infinite group and discrete/finite groups this integral becomes a sum. Although this is obvious, I feel like the authors should mention this point as the first G-Conv paper was on finite groups. With regards to Fig 1 & 2, I tried my very best but I still couldn't discern what to look for. I appreciate the authors outlining the important samples in red but still it wasn't clear to me. I feel like a better and fine grained qualitative description is needed here as at the moment it takes too much effort to understand what should be a simple figure. Moreover, proposition 4.1 is not novel but the current presentation is a bit ambiguous on this point. I'm sure the authors will agree that this has been proven in other work (e.g. [2]) so it would be necessary to make this more explicit and cite the correct papers. Finally, with regards to the experiments a quantification of the error is needed (e.g. +/- with a statistical test) as this is standard protocol.

***Technical Issues:***
This paper has a few other technical issues. First, it seems like a large body of Steerable G-CNNs are missed. This is important as the equivariance is guaranteed and not in expectation like in computing the integral. Furthermore, from the experiments it is unclear if the authors used Steerable G-CNNs for their baselines. I suspect that this is not the case as the results do not match [3] for RotationMNIST and CIFAR10. Also, do note that in [3] the authors achieve a variable level of equivariance at different depths which is in line with partial equivariance as proposed here. Admittedly, these are not the same but a similar idea was attempted in [3] and proved useful this diminishes the novelty of this work. If the baselines are indeed not G-Steerable with mixed equivariance then I feel like this is a must have and is quite a relevant comparison. Lastly, I found the quantification of the equivariance error an interesting discussion worthy of the main paper. However, I do not get the main point here as breaking exact equivariance to the whole group is desirable which is the premise of this paper. What should a reader take away from this error quantification if the error is not a bug but a feature? Finally, if I understood correctly the paper claims that we are learning a distribution over a subset but it seems like it is always the uniform distribution and what we instead learn is the domains or support of this distribution over the group. Is there a case where it would be beneficial if the learned distribution is not uniform?


[1] Kondor, Risi, and Shubhendu Trivedi. "On the generalization of equivariance and convolution in neural networks to the action of compact groups." International Conference on Machine Learning. PMLR, 2018.

[2] Cohen, Taco, Mario Geiger, and Maurice Weiler. "A general theory of equivariant cnns on homogeneous spaces." arXiv preprint arXiv:1811.02017 (2018).

[3] Weiler, Maurice, and Gabriele Cesa. "General $ E (2) $-Equivariant Steerable CNNs." arXiv preprint arXiv:1911.08251 (2019).

**Summary Of The Paper:**

This paper introduces the notion of partial equivariances which are defined as being equivariant with respect to a subset of a group. The authors also propose to learn the domain of this subset on the fly as opposed to enforcing this prior to training. The authors validate their proposed model on simple datasets that require different degrees of equivariance such as RotationMNIST, CIFAR10, and CIFAR100.

**Summary Of The Review:**

Overall the paper is of high production quality but misses the mark on claimed novelty. The biggest potential weakness is the omission of G-Steerable CNNs as baselines which are quite relevant here. There are a few minor presentation issues but these can be fixed with a revision.

---

> ### Author Response · Authors · 2021-11-17
> **First response Reviewer xU2c**
>
> Dear reviewer xU2c,
>
> First of all, we would like to thank you very much for your thorough and insightful review. We sincerely appreciate the time you spent in evaluating our work, and very much appreciate your comments.
>
> Here we will answer to all of your questions, comments and concerns:
>
> **[Presentation Issues]**
>
> **Starting with presentation, I found the notation to be unclear or inconsistent at times. For example, in Eqn 3. $f$ as previously defined is a map from $G\rightarrow \mathbb{R}$ but $v \in \mathcal{X}$. The lifting operation as a result is bit confusing. I'm aware of the lifting operation as defined in Kondor & Trivedi 2018 [1], can you please comment whether this is the same thing being done here?** Thank you for pointing this out. We have now improved the readability of this section on the revision of our paper. Yes, it is the same operation. In practical terms, this lifting can be seen as computing a convolution with a filter bank of group-acted versions of a convolutional kernel defined in the input domain.
>
> **With regards to Fig 1 & 2, I tried my very best but I still couldn't discern what to look for. I appreciate the authors outlining the important samples in red but still it wasn't clear to me. I feel like a better and fine grained qualitative description is needed here as at the moment it takes too much effort to understand what should be a simple figure.** We have now modified the captions of the figures, and improved the Figures. We hope that now it is easy to understand what it is going on in there. If you still have comments in this respect, please let us know. We are more than happy to incorporate them.
>
> **Moreover, proposition 4.1 is not novel but the current presentation is a bit ambiguous on this point. I'm sure the authors will agree that this has been proven in other work (e.g. [2]) so it would be necessary to make this more explicit and cite the correct papers.** This is correct. We did not mean to present this as a contribution of our own, but rather aimed to present it as common knowledge. We have now cited [2], and made it clear that we are rewriting it because it is important for our analysis only.
>
> **Finally, with regards to the experiments a quantification of the error is needed (e.g. +/- with a statistical test) as this is standard protocol.** We will include this in our paper. We have not done so yet, because we were busy running several other experiments for this rebuttal (see our response to Reviewer 3sxh for details). Nicely, Partial G-CNNs also perform positively in these newly added experiments.
>
> **[Technical Issues]**
>
> **First, it seems like a large body of Steerable G-CNNs are missed. This is important as the equivariance is guaranteed and not in expectation like in computing the integral.** We agree that we did not explicitly included an analysis on steerable G-CNNs. However, we did cite several works with steerable CNNs: Worrall et al., 2017; Weiler et al., 2018a, Weiler et al., 2018b, Esteves et al., 2019a;b; 2020, Sosnovik et al., 2019.
>
> In order to make clear the inclusion and relevance of Steerable CNNs in this line of work, we have included the following modifications:
> 1. Included Weiler & Cesa (2019) in the introduction as motivation for the exploration of different layer-wise levels of equivariance.
> 2. We have specified in Sec. 3, that the **practical implementation of group convolution** considered here hold for networks with regular group representations.
> 3. We have included the following paragraph in the discussion section of our work:
> > **Partial equivariances for other group representations.** Our theory of learnable partial equivariances is directly applicable to methods using regular group representations, e.g., Cohen & Welling (2016); Romero & Cordonnier (2020); Hutchinson et al. (2021). An extension for other kind of group representations such as irreducible representations in steerable CNNs, e.g., (Worrall et al., 2017; Weiler et al., 2018b; Weiler & Cesa, 2019) is an interesting research direction, as irreducible representations allow for exact equivariance to continuous groups.
>
> We hope that this changes make clear the relevance of steerable CNNs for this line of work.
>
> **I suspect that this is not the case as the results do not match [3] for RotationMNIST and CIFAR10** We indeed do not use steerable CNNs. This is because our method relies on the sampling of group elements to learn the distributions over the group. We believe that using steerable CNNs in this partial equivariant framework is an interesting direction for future work.

---

> > ### Author Response · Authors · 2021-11-17
> > **First response Reviewer xU2c -- continuation --**
> >
> > **Admittedly, these are not the same but a similar idea was attempted in [3] and proved useful this diminishes the novelty of this work** It is true that [3] uses different levels of equivariance at different layers. However, we note that they do **not** learn these during training, but instead, manually tune levels of equivariance at different layers to demonstrate that pure equivariance might be to restrictive in some settings. Note that manually tuning levels of equivariance in a network is a hard and time-consuming task, because it is not straightforward to define where to reduce equivariance, and by how much.
> >
> > Our proposed Partial G-CNNs, on the other hand, learn these partial equivariances directly from data. In addition, [3] transitions from groups to subgroups. Partial G-CNNs, on the other hand, also consider subsets of the group which themselves are not restricted to be a group.
> >
> > In addition, we would like to emphasize that learning levels of equivariance directly from data is non-trivial, and is in fact a hot area of research.
> >
> > **Lastly, I found the quantification of the equivariance error an interesting discussion worthy of the main paper.** We will try to include the main results from this Appendix in the main text.
> >
> > **However, I do not get the main point here as breaking exact equivariance to the whole group is desirable which is the premise of this paper. What should a reader take away from this error quantification if the error is not a bug but a feature?** Very good point. We observed that using the term "equivariance error" is not appropriate here, because error has an intrinsic negative connotation. We have now replaced this by the term *equivariance difference*, and have rewritten the corresponding sections to emphasize that this is not a bug but a feature, and that, in fact, this difference is what allows the network to break equivariance whenever necessary. Thank you for pointing this out.
> >
> > **Finally, if I understood correctly the paper claims that we are learning a distribution over a subset but it seems like it is always the uniform distribution and what we instead learn is the domains or support of this distribution over the group.** Your interpretation is correct. Note however, that learning the support of the distribution is equivalent to defining a distribution over the entire group, e.g., $[-\pi,\pi]$, and defining it to be zero outside of a learned region $[-\theta, \theta]$. In other words, it is equivalent to defining a uniform distribution on the entire group, which is zero outside of this learned region. This formality is important, because it ensures that the probability distribution can, if required, be defined on the entire group.
> >
> > **Is there a case where it would be beneficial if the learned distribution is not uniform?** We consider an uniform distribution in all our experiments as depicted in the previous response. However, there might exist some settings in which other kind of distributions are of interest. For instance, if we would like to transition from SO(2) to a cyclic group (e.g., rotations by 90 degrees), we would require a non-compact distribution. Although not considered empirically in this work, we provide a general definition of the probability distribution on the group to accommodate for other possible cases.
> >
> > **[Final words]** We hope that these responses clarify your questions and concerns. We will reflect this in our manuscript by the end of this week. Please do let us know if you have any follow-up / additional questions.
> >
> > Best regards,
> >
> > The Authors

---

> > > ### Comment · Reviewer_xU2c · 2021-11-18
> > > **Rebuttal response**
> > >
> > > I thank the authors for their time in responding to my questions and for any additional experiments that they might be running. A few of concerns still remain though. On a minor point, the notation in Eqn 3 still remains confusing. Specifically, $f: G \to \mathbb{R}$ but $v \in \mathcal{X}$. I apologize, if I'm mistaken here but this does not type check for me.
> > >
> > > With regards to the experiments. I will disagree with the authors here on the need for comparison with Steerable G-CNNs. The authors claim
> > >
> > > - "Note that manually tuning levels of equivariance in a network is a hard and time-consuming task, because it is not straightforward to define where to reduce equivariance, and by how much."
> > >
> > > But it seems atleast for the datasets considered this was not as time consuming and or challenging. Can you point to a specific new experiment that you have done which clearly demonstrates or somehow quantifies how much easier your approach is? Also please note that one can have Steerable G-CNNs with regular representations and not just irreducible ones. As a result, without this comparison I cannot see tremendous empirical value in this work at the moment.
> > >
> > > The ideas on the other hand are interesting and elegant but the authors could do more to find convincing use cases/domains/datasets where the proposed approach is clearly superior. I believe the community would benefit from this insight and fine grained comparison to Steerable G-CNNs. As a result, I will currently maintain my current score unless the authors are able to convincingly rebut this point.

---

> > > > ### Author Response · Authors · 2021-11-19
> > > > **Second response Reviewer xU2c**
> > > >
> > > > Dear Reviewer xU2c,
> > > >
> > > > thank you very much for your response.
> > > >
> > > > **On a minor point, the notation in Eqn 3 still remains confusing. Specifically,  $f: G \rightarrow \mathbb{R}$ but $v \in \mathcal{X}$. I apologize, if I'm mistaken here but this does not type check for me.** We have now changed the text before equation 3 to explain what is going on in there. *"Normally, the input of a neural network is not readily defined on the group of interest and thus group convolutions cannot be used directly. To do so, we first require an operation that lifts the input signal from its original domain $\mathcal{X}$ to the domain on which the group convolution is defined: the group $\mathcal{G}$. This operation is called a lifting convolution and is defined as:"*
> > > >
> > > > Please let us know if this is still unclear.
> > > >
> > > > **With regards to the experiments. I will disagree with the authors here on the need for comparison with Steerable G-CNNs. The authors claim "Note that manually tuning levels of equivariance in a network is a hard and time-consuming task, because it is not straightforward to define where to reduce equivariance, and by how much."But it seems at least for the datasets considered this was not as time consuming and or challenging.** We would like to emphasize that this is a difficult and time consuming task. To clarify this, we would like to refer you to the paper of Weiler & Cesa (2019).
> > > >
> > > > In their Tab. 4, you can observe that a sweep needs to be done in order to select a good performing "drop in equivariance" in the network. Note that the authors vary in this table not only the layer at which the equivariance drop happens, but also from which group to which subgroup it drops.
> > > >
> > > > This illustrates that where and to which subgroup to drop equivariance are **hyperparameters** of the model, which must be tuned manually. The same is illustrated for their models in Table 6. Here, once again, the authors perform a sweep over multiple possible values and dropping schemes.
> > > >
> > > > Note that these experiments are not exhaustive, and many more combinations could be possible. In contrast our models **learn** these drops in equivariance directly from data. Not only where, but also to which subset to drop.
> > > >
> > > > **Can you point to a specific new experiment that you have done which clearly demonstrates or somehow quantifies how much easier your approach is?** Please note from the previous response that all our experiments are actually showing improvements.
> > > >
> > > > In particular, the improvements come from **not** requiring to tune these hyperparameters manually, but letting the model learn them by itself.
> > > >
> > > > As a side node, we would like to note that our Partial G-CNNs achieve state of the art in rotMNIST, outperforming the models in Weiler & Cesa (2019). Nevertheless, we considered this to be irrelevant for the main contribution of our paper, which is that it is possible to learn equivariances and partial equivariances from data.
> > > >
> > > > With that being said, if you have any particular comparisons in mind, we are more than happy to include them in our paper.
> > > >
> > > > **Also please note that one can have Steerable G-CNNs with regular representations and not just irreducible ones.** Sorry if we were not clear about this in the paper. What we meant to say is that steerable G-CNNs must rely on irreducible representations to provide equivariance to continuous groups. Although steerable G-CNNs with regular representations can be constructed, this is only the case for discrete groups. This can, for instance, be seen in Table 3 of Weiler & Cesa (2019), where results for continuous groups are only presented for steerable G-CNNs irreducible representations.
> > > >
> > > > We will clarify this in our paper.
> > > >
> > > > Best regards,
> > > >
> > > > The authors.

---

> > > > ### Comment · Area_Chair_VqgP · 2021-11-26
> > > > **Please post your further opinions**
> > > >
> > > > Dear Reviewer xU2c,
> > > >
> > > > The authors have posted their rebuttal again. I wonder whether the rebuttal addressed your concerns? Please post your further opinions. Thanks!
> > > >
> > > > AC

---

### Official Review · Reviewer_8mxH · 2021-10-29

**Correctness:** 3
**Technical Novelty And Significance:** 3
**Empirical Novelty And Significance:** 2
**Recommendation:** 5
**Confidence:** 3

**Main Review:**

## Strengths
  1. The writing is clear and easy to follow. The relationship with relevant works is discussed sufficiently.
  2. The proposed partial equivariance is novel and several different cases of the subsets are properly discussed (e.g., group to subset, subset to group, etc.)
  3. Experiments show certain improvements compared with enforcing equivariance on the whole group.

## Weaknesses
  1. One major concern is that the partial equivariance seems not motivated properly. It is not clear why and when the partial equivariance will be needed in the natural datasets. All experiments on MNIST are artificially constructed in favor of the partial equivariance, and the experiment on CIFAR only show marginal improvement on classification accuracy which cannot prove that the CIFAR dataset demonstrates partial equivariance property. More analysis/visualization/experiments/concrete examples should have been included to motivate the need for partial equivariance and show its specific benefits.
  2. Another concern is related to the first one: The experiments are conducted only on small datasets and are artificially constructed in favor of the partial equivariance. More experiments on larger and natural datasets are needed.

**Summary Of The Paper:**

This paper proposes to learn partial equivariance where the equivariance property is constrained to a subset of the considered group. Specifically, they propose to learn such subset instead of predefining it, by parameterizing the sampling probability of the MC sampling. Experiments on variants of MNIST and CIFAR10 demonstrate that the partial equivariance brings better performance than whole group equivariance in terms of classification accuracy.

**Summary Of The Review:**

In summary, although the proposed partial equivariance is novel, it's not well motivated and the current experiments/analysis do not prove the benefits of partial equivariance.

---

> ### Author Response · Authors · 2021-11-17
> **First response Reviewer 8mxH**
>
> Dear reviewer 8mxH,
>
> First of all, we would like to thank you very much for your thorough and insightful review. We sincerely appreciate the time you spend in evaluating our work, and very much appreciate your comments.
>
> Here we will answer to all of your questions, comments and concerns:
>
> **One major concern is that the partial equivariance seems not motivated properly. It is not clear why and when the partial equivariance will be needed in the natural datasets.** We apologize if this is not clear in the paper. We aimed to explain the importance of partial equivariance in our introduction with a running example:
>
> >... However, the group to which G-CNNs are equivariant must be fixed prior to training, and imposing equivariance to a group larger than the symmetries present in the data leads to over-constraining and worse performance (Chen et al., 2020). The latter results from a difference in the distribution of the data, and the family of distributions the model can describe.
> Consequently, the group must be selected carefully, and it should correspond to the transformations that appear naturally in the data.
>
> > Interestingly, the distribution of the data can frequently be better represented by a subset of the group than by the group itself, e.g., rotations in [−90○ , 90○]. For instance, natural images are much more likely to show an elephant standing straight or slightly rotated than an elephant upside-down. In some cases, group transformations can even change the desired model response. A typical example is the classification of the numbers 6 and 9, whose defining factor is their pose. In both examples, the distribution of the data is better represented by a model that respects rotation equivariance partially. That is, a model equivariant to some, but not all rotations.
>
> > In addition, the optimal family of full / partial equivariances may change per layer. This results from changes in pose likelihoods for low and high-level features in the data, e.g., the orientations of edges in an human face, and the pose of human faces relative to the camera. Whereas detecting a face may benefit from partial rotation equivariance, detecting edges benefits from full rotation equivariance.
>
> > These observations indicate that constructing a model with different levels of partial / full equivariances at each layer may be advantageous. Weiler & Cesa (2019) empirically show that manually selecting the level of equivariance at different layers can lead to accuracy improvements. However, manually defining layer-wise levels of equivariance is at best a difficult and time-consuming task.
>
> With this running example we aim to explain that several objects in real images will have a preferred orientation. For instance, the probability of seeing a plane or a ship upside down is very low in comparison with the probability of seeing them straight. Consequently, a model could benefit from learning to be equivariant to the poses that normally appear in data.
>
> In addition, since edges and other low-level features still can appear at all positions regardless of the general pose of the object, the optimal level of equivariance might differ per layer.
>
> We are more than happy to include any additional ideas you have in this respect.
>
> **All experiments on MNIST are artificially constructed in favor of the partial equivariance.** Please note that if all the datasets were exactly equivariant, then the network would not need to learn partial equivariances. This is why we evaluate the capacity of our models to learn partial equivariances. Note however, that we also use rotMNIST in our evaluation: a dataset broadly used to evaluate full equivariance. Importantly, partial G-CNNs learn to stay fully equivariant if this is advantageous for the task at hand, and match conventional G-CNNs in this setting.
>
> In other words, partial G-CNNs can be fully equviariant if required, but can learn to reduce their equivariance if this is beneficial.

---

> > ### Author Response · Authors · 2021-11-17
> > **First response Reviewer 8mxH -- continuation --**
> >
> > **More analysis/visualization/experiments/concrete examples should have been included to motivate the need for partial equivariance and show its specific benefits. The experiments are conducted only on small datasets and are artificially constructed in favor of the partial equivariance. More experiments on larger and natural datasets are needed.** We have included additional experiments. In particular, we replace the final layer of a Partial G-CNN with a T2 layer, and the max pooling operation at the end with a learnable MLP. (Please see our answer to Reviewer 3sxh for these results).
> >
> > Our results demonstrate that Partial G-CNNs perform better than these baselines. This illustrates that learning layer-wise equivariances is advantageous for CIFAR10 and CIFAR100. If you would like to have any other experiments in mind. Please let us know. We are more than happy to include them in our work.
> >
> > **[Final words]** We hope that these responses clarify your questions and concerns. We will reflect this in our manuscript by the end of this week. Please do let us know if you have any follow-up / additional questions.
> >
> > Best regards,
> >
> > The Authors

---

> ### Comment · Area_Chair_VqgP · 2021-11-26
> **Please respond to the author rebuttal**
>
> Dear Reviewer 8mxH,
> The authors have posted their rebuttal. I wonder whether the rebuttal addressed your concerns? Please respond to the authors. Thanks!
>
> AC

---

> > ### Comment · Reviewer_8mxH · 2021-11-26
> > **Respond to the author rebuttal**
> >
> > Dear authors and AC,
> >
> > I have read the authors' rebuttal and appreciate the authors' efforts. I agree that it is an interesting and valuable research direction. However, I'm still not fully convinced about the "partial equivariance" part. The current results only show (marginal) improvements in terms of accuracy numbers compared with baseline methods, but there are no visualizations or specifically designed experiments to show that the network is indeed learning "partial equivariance", what kind of subset it is learning to be equivariant to, and why this learned subset helps boost the performance. From this perspective, I would like to maintain my current score.

---

> > > ### Author Response · Authors · 2021-11-29
> > > **Response**
> > >
> > > Dear reviewer,
> > >
> > > thank you very much for your answer.
> > >
> > > **The current results only show (marginal) improvements in terms of accuracy numbers compared with baseline methods, but there are no visualizations or specifically designed experiments to show that the network is indeed learning "partial equivariance", what kind of subset it is learning to be equivariant to, and why this learned subset helps boost the performance.** We have indeed not included all of these results in our paper. But we intend to. For more information on these results, as well as the learned subsets and several other experiments (comparisons with other baseliens as well as Augerino (Benton et al),  we encourage you to take a look at our response to Reviewer 3sxh (the one above yours). We believe that our answer to his review will give clarity to the problem at hand.
> > >
> > > Best regards,
> > >
> > > The authors

---

### Official Review · Reviewer_3sxh · 2021-11-02

**Correctness:** 3
**Technical Novelty And Significance:** 3
**Empirical Novelty And Significance:** 2
**Recommendation:** 5
**Confidence:** 4

**Main Review:**

# Strengths

The paper addresses the important problem of learning symmetries from data. One of the weaknesses of group equivariant CNNs is that one has to assume what is the group that acts on the dataset, which might be hard to know in practice. Commonly, the chosen group will be too restrictive and still miss relevant transformations. The paper proposes a way to start from some group of transformations and learn to prune it differently at each layer. This is potentially more flexible and is shown to outperform the baselines where the original group is fixed throughout.

# Weaknesses

If I understand correctly, the partial equivariance as defined is an approximate equivariance to a subset of the group. In this case, the average along the group dimension will not be invariant to transformations from the subset. Furthermore, once equivariance is broken in some layer, it can never be recovered, so the effect of this approach is to monotonically restrict equivariance on each subsequent layer.

On one hand, this is satisfying because the benefits of reducing or breaking equivariance at some mid or final layer have been observed before (eg the group restriction in Weiler and Cesa [1]). On the other hand, what the method can learn from data seems to be straightforward to implement with either a group restriction or including coordinate values on feature channels like in Fuchs et al [2], while the proposal seems also prone to converging to undesirable models such as breaking equivariance too early or having a more restrictive layer following a less restrictive.

1) Figure 4 shows what I would expect as a reasonable output, where the equivariance is broken at the final layers to make use of the input pose. Is the same observed for CIFAR10/100? Please report similar figures for the SE(2) case.

2) Results would be more convincing if compared with the aforementioned handcrafted methods of breaking equivariance. How would something simple like dropping from E(2) to T(2) on the last layer, or using a fully connected layer instead of the final global max pooling perform on CIFAR10/100? These would highlight the advantages learning partial equivariances per layer.

3) The Monte Carlo approximation of the group convolution integral causes some equivariance error. The group pruning to achieve partial equivariance compounds on this error, even when only the group subset is considered. I appreciate the error expressions in Appendix B, but I believe such errors should be quantified and compared with the baselines (both non-equivariant and fully equivariant). Without these numbers it is hard to judge whether (or at what layer) we can consider the equivariance approximate or broken.

I have a separate concern regarding novelty. The paper seems heavily inspired by Benton et al [3], who, despite focusing on learning invariances from data, also show an extension to learning global equivariances. I believe the differences and similarities between both approaches should be discussed more thoroughly and that quantitative comparison under fair conditions is also necessary. Benton et al [3] also show results for rotated MNIST and CIFAR10.

# Questions

1) Please elaborate on which discrete groups are unstable to train as mentioned in section 6. Table 2 shows the best results for E(2) G-CNN, which has a discrete component. Was the instability observed in this model or others?

# Minor

Figures 1 and 2 are hard to read. It would be much easier to compare the feature maps and see the differences if they were rotated to the same pose.

# References

[1] Weiler and Cesa, "General E(2) - Equivariant Steerable CNNs", NeurIPS'19.

[2] Fuchs et al, "SE(3)-Transformers: 3D Roto-Translation Equivariant Attention Networks", NeurIPS'20.

[3] Benton et al, "Learning Invariances in Neural Networks", NeurIPS'20.

[4] Finzi et al, "Generalizing Convolutional Neural Networks for Equivariance to Lie Groups on Arbitrary Continuous Data", ICML'20.


**Summary Of The Paper:**

The paper proposes to learn partial equivariances from data by training a group equivariant CNN (G-CNN) where the group elements are pruned at each layer. For learning the pruning operation, the paper leverages the idea of Finzi et al [4] to approximate the group convolution with a Monte Carlo integral, and the idea of Benton et al [3] to use the reparametrization trick to learn the bounds of an uniform distribution from where the group elements are sampled. Results show that the partial equivariance performs outperforms the full equivariance when the data itself is not fully rotational-equivariant.

**Summary Of The Review:**

I think the paper addresses an important problem and proposes a seemingly reasonable method that might be useful in practice. However, in the current state I'm not sure if there is enough evidence to demonstrate the value of the paper (see detailed review). I would be happy to increase my rating if such evidence is presented during the discussion phase.

---

> ### Author Response · Authors · 2021-11-17
> **First response Reviewer 3sxh**
>
> Dear reviewer 3sxh,
>
> First of all, we would like to thank you very much for your thorough and insightful review. We sincerely appreciate the time you spent in evaluating our work, and very much appreciate your comments.
>
> Here we will answer to all of your questions, comments and concerns:
>
> **[Summary of the paper]**
>
> **... and the idea of Benton et al [3] to use the reparametrization trick to learn the bounds of an uniform distribution from where the group elements are sampled.** We note that in addition to the reparameterization trick used by Benton et al to learn distributions over continuous transformations, we also define a way to learn distributions over discrete transformations. This is not possible with the reparameterization used in Benton et al. We have made this clear in the new revision of our work.
>
> **[Weaknesses]**
> **The partial equivariance as defined is an approximate equivariance to a subset of the group. In this case, the average along the group dimension will not be invariant to transformations from the subset.** I am not sure I understand this statement. We would like to emphasize that breaking equivariance is exactly what we want in order for the model to be able to detect changes in transformations of the input, and how much equivariance is broken is controlled by the learned group subsets.
>
> In other words, that equivariance can be broken is not a bug, but a feature. In order to make this clearer, we have modified multiple parts of our manuscript, and stopped referring to the equivariance change as equivariance error (which has a negative connotation), and refer to it now as "equivariance difference".
>
> **Furthermore, once equivariance is broken in some layer, it can never be recovered, so the effect of this approach is to monotonically restrict equivariance on each subsequent layer.** This is not necessary. We do not require to impose a monotonic decrease in equivariance. Note however, that, although once broken, equivariance cannot be recovered, the equivariance error is controlled by the size of the input and output subsets at every layer. As a consequence, even if equivariance breaks in the middle of the network, by considering fully equivariant layers after that, the equivariance difference is kept equal from there onwards.
>
> To demonstrate that equivariance does not need to be monotonically decreasing in our model, we include an additional monotonic equivariance loss to our training given by:
>
> $$L_{\text{mon.~equiv}} = \sum_{l=1}^{L-1}(\gamma_l - \max(\gamma_{l+1}, \gamma_l))$$
>
> where $\gamma_l$ represents the current limit of the subset learned at the layer $l$. In other words, we penalize the model if the subset at the subsequent layer is larger than the previous layer.
>
> Our results are:
>
> | Group | # Elements | rotMNIST | CIFAR10 | CIFAR100 |
> | --- | --- | ---| ---| --- |
> | SO(2) | 16 |  99.15 | 87.02 | 57.11 |
> | O(2) | 16 | 98.41 | 89.00 | 58.85 |
>
> In other words, imposing a monotonic equviariance decrease leads to slightly worse performance than an unconstrained model. We will complete these experiments for 8 elements at each group, and will add this experiments to a "Ablation studies" section in the Appendix.

---

> > ### Author Response · Authors · 2021-11-17
> > **First response Reviewer 3sxh -- continuation --**
> >
> > **...is satisfying because the benefits of reducing or breaking equivariance at some mid or final layer have been observed before (eg the group restriction in Weiler and Cesa [1]).** This is true. In fact we were partially motivated from their observation. We apologize for not acknowledging the authors in a sufficient manner. This has been fixed now in the introduction of our paper.
> >
> > **On the other hand, what the method can learn from data seems to be straightforward to implement with either a group restriction...** We respectfully disagree with this comment. Note that [1] does not learn where and by how much to restrict equivariance. In fact, they manually decide where to reduce the level of equivariance of the model. In addition, [1] does not consider the setting of partial equivariances. Whenever they drop equivariance, they do so by selecting a subgroup of the previous layer.
> >
> > We note that selecting a layer on which to reduce equivariance is not straightforward. Similarly, it is also not straightforward to deduce the level on which equviariance must be reduced. For instance, should we go from O(16) to O(8) or O(4) at a certain layer? what about going to SO(16), SO(8) or SO(4) instead? At which layer should we do which?
> >
> > This problem is exacerbated if we do not consider subgroups only. As mentioned by you in the **strenghts** section, "One of the waknesses of group equivariant CNNs is that one has to assume what is the group that acts on the dataset, which might be hart do know in practice". Note that this becomes a much more difficult problem if we must do so at each layer.
> >
> > **...while the proposal seems also prone to converging to undesirable models such as breaking equivariance too early or having a more restrictive layer following a less restrictive.** We respectfully disagree that these models are undesirable. If this were the case, then the performance of these models would also be worse. Note that this is not the case either in our main experiments as well as in the additional experiments included in this rebuttal. That the model can learn to be restrictive at some middle layer rather indicates that selecting this manually is a hard task.
> >
> > **Figure 4 shows what I would expect as a reasonable output, where the equivariance is broken at the final layers to make use of the input pose. Is the same observed for CIFAR10/100? Please report similar figures for the SE(2) case.** We will include these results in the paper. We note that our models are not constrained to be monotonically decreasing in terms of equivariance. Consequently, the optimal subsets do not need to follow the structure shown in Fig. 4.
> >
> > For completeness, we provide here the learned subsets for SE(2)-CNNs with 16 elements:
> >
> > **rotMNIST**:
> > > $[-\pi, \pi)$ * tensor([1.0610])
> > tensor([0.9312])
> > tensor([1.0527])
> > tensor([1.0564])
> > tensor([1.0565])
> >
> > **CIFAR10**:
> > > $[-\pi, \pi)$ * tensor([0.9601])
> > tensor([0.9709])
> > tensor([0.8271])
> > tensor([0.7383])
> > tensor([0.7610])
> >
> > **CIFAR-100**:
> > > $[-\pi, \pi)$ * tensor([0.8814])
> > tensor([0.9353])
> > tensor([1.0419])
> > tensor([1.0135])
> > tensor([1.0063])
> >
> > We emphasize that Partial G-CNNs consistently outperform non-equivariant and fully equivariant ones if fully equivariant models are not optimal, and are equally performing to fully equivariant models otherwise.
> >
> > **Results would be more convincing if compared with the aforementioned handcrafted methods of breaking equivariance. How would something simple like dropping from E(2) to T(2) on the last layer, or using a fully connected layer instead of the final global max pooling perform on CIFAR10/100? These would highlight the advantages learning partial equivariances per layer.** Thank you very much for your suggestion. This is indeed an interesting experiment and will be included in the paper under the Ablation studies section. Our results are as follows:
> >
> > *Fully equivariant network with T2 layer at the end instead of group convolution*:
> >
> > |Group |	# Elements |	rotMNIST |	CIFAR10 |	CIFAR100 |
> > | --- | --- | --- | --- | --- |
> > |SO(2) |	16 |	99.04 |	82.76  |	52.51 |
> > |O(2)  |	16 |	97.98 |	86.68 | 57.61 |
> >
> > *Fully equivariant network with MLP instead of  layer at the end*:
> >
> > |Group |	# Elements |	rotMNIST |	CIFAR10 |	CIFAR100 |
> > | --- | --- | --- | --- | --- |
> > |SO(2) |	16 |	99.00 |	86.25  |	56.29 |
> > |O(2)  |	16 |	99.02 |	87.43 | 58.87 |
> >
> > In particular, we observe that simply replacing the last layer by a T2 layer does not perform particularly well. Interestingly, however, it seems that replacing max pooling by an MLP at the end of the network can be used to learn to pay attention to certain parts of the group positions during the calculation of the final feature maps. Nevertheless, we observe that Partial G-CNNs still perform better than both variants.
> >
> > We agree that these results will strengthen the position of our paper and most sincerely thank you for these suggestions.

---

> > > ### Author Response · Authors · 2021-11-17
> > > **First response Reviewer 3sxh -- continuation --**
> > >
> > > **I appreciate the error expressions in Appendix B, but I believe such errors should be quantified and compared with the baselines (both non-equivariant and fully equivariant).** Note that the derivations in Appx. B do not consider the equivariance error resulting from Monte Carlo approximations. As such, these results exactly quantify the equivariance difference resulting from partial equivariances. Do you have any particular experiment in mind which you would like to see for this particular point?
> > >
> > > **...it is hard to judge whether (or at what layer) we can consider the equivariance approximate or broken.** We can derive how much equivariance change at each layer based on the learned subsets (we will provide tthe learned subsets for all models in the paper). I must say, however, that I do not exactly understand what you mean here with "equivariance approximate or broken".
> > >
> > > Do you mean with approximate equivariance whether the model is fully equivariant but presents errors due to the Monte Carlo approximation, and with broken equivariance if the model is learning partial equivariances? If this is the case, we can obtain this information from the learned subsets. If the value of the subset is $[-\pi, \pi)$, then any equivariance error is only a result of the Monte Carlo approximation. If the learned subset is, for instance $[-0.5\pi, 0.5\pi)$, then there is a clear difference in equivariance, which can be computed for particular input transformations by relying on the terms provided in Appx. B.
> > >
> > > Please let us know if there is something else we can clarify in this regard. We are more than happy to do so.
> > >
> > > **I have a separate concern regarding novelty. The paper seems heavily inspired by Benton et al [3], who, despite focusing on learning invariances from data, also show an extension to learning global equivariances.** We have extended our discussion to handle more differences between Benton et al and our work. With regard to their experiments regarding global equivariances, we note that this is only possible in their framework for models for which the output also can exhibit equivariance, e.g., segmentation. As a result, equivariant models for classification cannot be learned with the Augerino framework.
> > >
> > > Furtermore, we include a method to learn equivariance to discrete transformations (e.g., mirroring). This cannot be accommodated within the Augerino framework, which relies on the reparameterization trick.
> > >
> > > In addition, our model learns partial equivariances at every layer of the network. To the best of our knowledge, there is no other method in the literature that proposes this. Moreover, we explicitly provide an interpretation of our framework in terms of groups and group equivariance. Furthermore, we provide the exact difference in terms of equivariance as a function of the learned subsets.
> > >
> > > We believe that these contributions illustrate the novelty of our work.
> > >
> > > **...and that quantitative comparison under fair conditions is also necessary.** We are currently working on this comparison. We hope to have them ready before the end of this rebuttal period.
> > >
> > > **[Questions]**
> > >
> > > **Please elaborate on which discrete groups are unstable to train as mentioned in section 6. Table 2 shows the best results for E(2) G-CNN, which has a discrete component. Was the instability observed in this model or others?** Exactly, this is the discrete component we refer to in the paper. We will clarify this in the text.
> > >
> > > **[Minor]**
> > >
> > > **Figures 1 and 2 are hard to read. It would be much easier to compare the feature maps and see the differences if they were rotated to the same pose.** We have now improved the images as well as the captions. Any comments regarding this will be much appreciated.
> > >
> > > **[Final words]** We hope that these responses clarify your questions and concerns. We will reflect this in our manuscript by the end of this week. Please do let us know if you have any follow-up / additional questions.
> > >
> > > Best regards,
> > >
> > > The Authors

---

> > > > ### Author Response · Authors · 2021-11-23
> > > > **Results Augerino comparisson**
> > > >
> > > > Dear reviewer 3sxh,
> > > >
> > > > we finally have results for our comparison with Augerino (Benton et al. 2020). For Augerino, we sample # elements samples from the group, augment the input with each of the sampled transformations, and pass them through the network. Next, we take the mean of the predictions, and use it as the final prediction.
> > > >
> > > > Note that Augerino only works with continuous groups. By using our proposed distribution parameterization, we extend it to include discrete transformations as well (E2). The results are as follows,where the results of Partial G-CNNs have been included in parenthesis at each row to ease comparison:
> > > >
> > > > | Group	|# Elements	|rotMNIST	|CIFAR10	|CIFAR100|
> > > > |---|---|---|---|---|
> > > > |SE(2)	|8	|99.17 (99,.23)	|82.38 (88.59)	|52.98 (57.26)|
> > > > |SE(2)	|16	|99.25 (99.18)	| 82.53 (88.59) | 51.47 (57.31) |
> > > > |E(2)	|8	|99.12 (97.78)	| 84.29 (89.00)	|52.59 (55.22) |
> > > > |E(2)	|16	|99.18 (98.35)	| 83.54 (90.12)	| 54.76 (61.46) |
> > > >
> > > > **Analysis**. On rotMNIST, Augerino performs very well and pairs partial G-CNNs. Rotated MNIST is constructed by applying global rotations to the MNIST dataset.  Consequently, the good performance of Augerino can be explained given the nature of the dataset.
> > > >
> > > > For CIFAR-10 and CIFAR-100, however, we observe that there is a clear difference between Augerino and partial G-CNNs, with partial G-CNNs performing much better than Augerino. We believe that this comes through the fact that local changes in pose play a much more important role in this dataset than for rotMNIST, where MNIST images are globally modified. Consequently, tackling different levels of equivariance at each layer leads to benefits upon tackling a single level of global invariance for the entire network.
> > > >
> > > > Best regards,
> > > >
> > > > The authors.

---

> ### Comment · Area_Chair_VqgP · 2021-11-26
> **Please respond to the author rebuttal**
>
> Dear Reviewer 3sxh,
> The authors have posted a looong rebuttal to your comments. I wonder whether the rebuttal addressed your concerns? Please respond to the authors. Thanks!
>
> AC

---

> ### Comment · Reviewer_3sxh · 2021-11-27
> **Response to rebuttal**
>
> Thank you for the thorough response and new experiments.
>
> I think my points about "breaking equivariance" were misunderstood. I expected "partial equivariance" to a group to mean that when applying transformations from a subset of the group to the inputs, the feature maps would undergo the same transformation. While this seems to be true for a single layer, it is not true when stacking many layers. So in that sense the model could be no more equivariant (in terms of an equivariance error $\mid f(\text{rotated}(x))- \text{rotated}(f(x))\mid$) than a standard neural network. Hence my suggested experiment of evaluating the equivariance errors by measuring the distance between rotated  inputs and feature maps and comparing with both fully equivariant and non-equivariant methods.
>
> In fact the experiments are mostly on uniformly rotated datasets like rotMNIST, where the network must only learn to do nothing and keep the original internals, or upright datasets where no level of global equivariance is required. It would be interesting to see what happens with some actual "partial symmetries" such as the datasets in Benton et al where rotations are constrained to a small interval or union of disjoint intervals -- would the trained model exhibit the correct partial equivariance as measured by the equivariance error on the last layer?
>
> I realize it is unfair to ask for new experiments this close to the deadline, but these are somewhat minor points; my concerns about novelty still persist and I intend to maintain my score.

---

> > ### Author Response · Authors · 2021-11-29
> > **Response on Rebuttal**
> >
> > Dear reviewer,
> >
> > Thank you very much for your response.
> >
> > **I think my points about "breaking equivariance" were misunderstood. I expected "partial equivariance" to a group to mean that when applying transformations from a subset of the group to the inputs, the feature maps would undergo the same transformation. While this seems to be true for a single layer, it is not true when stacking many layers. So in that sense the model could be no more equivariant (in terms of an equivariance error $\| f(\text{rotated}(x)) - \text{rotated}(f(x)) \|$ ) than a standard neural network. Hence my suggested experiment of evaluating the equivariance errors by measuring the distance between rotated inputs and feature maps and comparing with both fully equivariant and non-equivariant methods.** We are sorry to hear that this was misunderstood. However, we would like you to note that we have derived an analytic expression for the calculation of the equivariance error. Nevertheless, we agree that it is interesting to make sure this holds in practice. We can add this to the final version of our manuscript. However, we believe this does not change the core contribution of our paper and that not having it, does not diminish the relevance of our analytic error expression.
> >
> > **In fact the experiments are mostly on uniformly rotated datasets like rotMNIST, where the network must only learn to do nothing and keep the original internals, or upright datasets where no level of global equivariance is required.** We do not agree with this statement. We have provided evidence that our networks outperform fully equivariant networks as well as non-equivariant networks. We do not agree that we provide results only on datasets with full equivariance or "upright datasets where no level of global equivariance is required". We do not agree that CIFAR-10 / 100 do not exhibit any level of variation. It is true that these datsets are upright (in that no upside down samples are to be expected), but some level of transformations naturally appear on data. Note that if it were the case that CIFAR-10 / 100 are simply upright datasets, our Partial G-CNNs would not improve upon the results given by conventional CNNs and would not learn partial subsets of the groups (the learned subgroups have been provided in our answer to your original review.
> >
> > We are most surprised by your answer. We would like you to note that not only have we provided results in these datasets, but we have also provided an extended list of additional results that show the validity of our method. Based on your original review, we demonstrate that it is better to have Partial G-CNNs that a final T2 layer at the end, or even an MLP instead of global pooling. In addition, we show that Partial G-CNNs also outperform Augerino. This also illustrates that learning equivariances (as in our paper) is better than learning invariances (as in Benton et al). Note, additionally, that we also provide a way to learn distributions over discrete transformations.
> >
> > Furthermore, we have provided as answer to your rebuttal the sets of transformations that are being learned by Partial G-CNNs. As you can observe, it is not true that *"the network must only learn to do nothing and keep the original internals, or upright datasets where no level of global equivariance is required"*. As you can see, the network is indeed learning subsets of the group which notably leads to improvements in accuracy.
> >
> > **It would be interesting to see what happens with some actual "partial symmetries" such as the datasets in Benton et al where rotations are constrained to a small interval or union of disjoint intervals** We can add more experiments showing results in other toy examples as you propose. However, we would like you to note that we provide results on natural images and show that partial equivariances lead to improvements in performance. Consequently, although these additional toy experiments would be of added value from a didactic perspective, they would not change the current value of our experimental section.
> >
> > **I realize it is unfair to ask for new experiments this close to the deadline, but these are somewhat minor points; my concerns about novelty still persist.** We sincerely do not understand your concerns about novelty. We introduce a way to learn partial equivariances (not only is the concept of partial equivariances novel, but also how to learn them during training). In addition, not only partial G-CNNs work for continuous works (which is also novel) but we also provide a parameterization to learn equviariances on discrete groups (novel and not adressed by Benton et al, even in the invariance case). Our results, additionally show that Parital GCNNs outperform Augerino as well as several baseline networks.
> >
> > We hope this helps to illustrate the novelty and added value of our work.
> >
> > Best regards,
> >
> > The Authors

---

### Official Review · Reviewer_RivC · 2021-11-02

**Correctness:** 3
**Technical Novelty And Significance:** 4
**Empirical Novelty And Significance:** 4
**Recommendation:** 6
**Confidence:** 4

**Details Of Ethics Concerns:**

No ethics concerns

**Main Review:**

**Strengths**:

- Partial equivariance is an interesting yet under-explored problem. Existing G-CNN literature mainly focuses on full equivariance, and partial equivariance is seldom explored. This paper has demonstrated that studying partial equivariance is of practical relevance: It can improve performance upon conventional G-CNNs when full equivariance is harmful.
- The proposed partial group convolution is a simple yet effective solution. This paper proposes a solution that only requires a small modification of existing G-CNN architectures, and the authors have empirically shown that the solution is effective.
- Clear messages from the experiments. I like the experiments section, because the take-away messages from these experiments are very clear, and the main claims of this paper are empirically well supported:
  - (Figure 4) Different layers may require different levels of equivariance.
  - (Table 2) Partial G-CNNs improve performance when full equivariance is harmful, e.g. CIFAR10, CIFAR100, and still match the performance of G-CNNs when full equivariance is beneficial, e.g. rot-MNIST
  - (Table 4) SIREN kernel parameterisation is significantly better compared to other choices.

**Weaknesses**:

My main concern of this paper is its presentation and technical statements, which is detailed below:
- Misleading terminology/notations. Even though the overall idea is simple and clear to me. I still find the technical parts very hard to follow and check their correctness, mainly due to the somewhat misleading usage of terminology and notations. Below are a few examples:
  - The input/output space of group convolutions are functions on groups rather than the groups themselves, and the groups are the input space of a particular input/output function. This paper very often does not distinguish the two, hence causing confusion.
  - Does $p_i$ in Section 4.3.1 (the discrete group case) define a categorical distribution over groups? If that is the case, then the probabilities should sum to 1? But then the paper set all the probabilities $p_i$ to $1$. I might misunderstand something here.
- Unclear illustration: Figure 1 and Figure 2 are hard for me to understand. A few suggestions to improve clarity:
  - Provide a mathematical definition of equivariance error.
  - In Figure 1, images on the right are feature maps (functions on groups) rather than group elements. So better not annotate them as group elements and group subsets, or explain what’s the correspondence between group elements and feature maps?
Images in Figure 2 might be too small.
  - The text explanation for these two figures also needs lots of work.


**Questions**:

- If there is an image dataset, for some of the data, the corresponding group subset is $[-60, 60]$, for the rest, the corresponding group subset is $[0, 90]$, what would the overall learned group subset be?
- In the definition of partial group convolution, can we constrain the distribution over groups to be a uniform distribution over a subset? That is to say, group elements are either in or out of the subset, and all elements in the subset are equally likely. For me, it seems to be a more ‘natural’ definition of ‘partial’ group convolution, unless there are cases when you would prefer a more flexible distribution $p(g)$?

I will give a negative score for now because the technical part needs significant work before publishing. But I think it still solves an interesting problem with a simple solution, and the experiments are sufficient. If the authors are willing to give a list of changes they will make to the mathematical formulation and statements, I am happy to raise my score to the positive side.


**Summary Of The Paper:**

**Summary and Contributions**:

The paper proposes partial group convolution that can learn layer-wise partial equivariance from data. The main idea is to use the Monte Carlo approximation to the group convolution as in LieConv (Finzi 2020), but sample group elements from a learned distribution over groups rather than a uniform one.

They empirically show that allowing the network to learn partial equivariance improves performance upon conventional G-CNNs when full equivariance is harmful, and still match the performance of G-CNNs when full equivariance is beneficial.


**Summary Of The Review:**

The paper studies an interesting problem proposes a simple solution and has sufficient empirical support for main claims. However, the technique parts and over clarity needs significant work before publishing.

---

> ### Author Response · Authors · 2021-11-17
> **First response Reviewer RivC**
>
> Dear reviewer RivC,
>
> First of all, we would like to thank you very much for your thorough and insightful review. We sincerely appreciate the time you spent in evaluating our work, and very much appreciate your comments.
>
> Here we will answer to all of your questions, comments and concerns:
>
> **[Weaknesses]**
>
> **Misleading terminology/notations. Even though the overall idea is simple and clear to me. I still find the technical parts very hard to follow and check their correctness, mainly due to the somewhat misleading usage of terminology and notations. Below are a few examples:** You are indeed right. We apologize for these inconsistencies. We have now submitted an updated version of our work in which we filter out the misleading usage of terminology and notations.
>
> With regard to your particular examples:
>
> > The input/output space of group convolutions are functions on groups rather than the groups themselves, and the groups are the input space of a particular input/output function. This paper very often does not distinguish the two, hence causing confusion.
>
> We have now make clear what do we mean with space and domain. You are right. We often were referring to the domain of the input and output feature maps, but refer to them as the "input/output space". We have now corrected this.
>
> > Does $p_i$ in Section 4.3.1 (the discrete group case) define a categorical distribution over groups? If that is the case, then the probabilities should sum to 1? But then the paper set all the probabilities  $p_i$ to $1$. I might misunderstand something here.
>
> This section of our paper was indeed misleading and confusing. We have rewritten this part now, and hopefully it is easy to understand now.
>
> To learn a distribution on the discrete group, we would need to enumerate of possible combinations of the elements in the discrete group and assign a probability to each of this combinations. However, the number of combinations grows exponentially with the size of the group, which can lead to instabilities while learning the distribution. To overcome this issue, we instead parameterize this probability independently for each group element. That is, we define the probabilities of sampling a particular element or not. This strategy scales linearly with the number of elements in the group.
>
> In addition to these observations, we have improved the overall readability of our work. We thank you very much for your constructive comments in this regard.
>
> **Unclear illustration: Figure 1 and Figure 2 are hard for me to understand. A few suggestions to improve clarity** We have improved the figures and the captions of these images. If there are additional comments in this regard, we would be happy to hear them.
>
> **[Questions]**
>
> **If there is an image dataset, for some of the data, the corresponding group subset is $[-60, 60]$, for the rest, the corresponding group subset is $[0, 90]$, what would the overall learned group subset be?** This is a very interesting question. If I get the core of your question, you are suggesting that the subset should be dependent on each instance / class of a dataset and not equal for the entire dataset. If this assumption is correct, I entirely agree with you. This is one of the main drivers of our subsequent work of this paper.
>
> To answer your question: it depends. In the current framework, the model is not able to learn different subsets for different data samples. As a result, the model would need to try and do its best in fitting both requirements. If possible, I assume that the model would try to change the requirement $[0, 90]$ to $[-45, 45]$. If this is possible, I would expect the resulting subset to be somewhere between $[-45, 45]$. and $[-60, 60]$. Where exactly in between would be given by the relative number of samples in the dataset that require one subset or the other.
>
> If the conversion from $[0, 90]$, to $[-45, 45]$ is not possible (although I am not sure this would be the case, because canonical positions do not exist for group convolutional kernels), then I would expect the resulting subset to be between $[-60, 60]$ and $[0, 90]$. Similarly to the previous example, how close the subset would be to one or the other subset depends on the amount of samples in the dataset that require each of these subsets.
>
> It is important to mention, however, that this is purely a thought exercise. In practice, our learned distributions on SO(2) are always centered around zero, and thus, learning a subset $[0, 90]$ is not possible.

---

> > ### Author Response · Authors · 2021-11-17
> > **First response Reviewer RivC -- continuation --**
> >
> > **In the definition of partial group convolution, can we constrain the distribution over groups to be a uniform distribution over a subset? That is to say, group elements are either in or out of the subset, and all elements in the subset are equally likely. For me, it seems to be a more ‘natural’ definition of ‘partial’ group convolution, unless there are cases when you would prefer a more flexible distribution ?** This is a good observation. This is exactly what we do.
> >
> > As mentioned in Sec. 4.3.1., we learn the "limits" in which the uniform distribution is non-zero, e.g., $\theta$ for SO(2). Observe however, that having a distribution that is uniform over some range, e.g., $[-\theta, \theta]$, is equivalent to defining a distribution which is uniform over that range and zero everywhere else, e.g., uniform if $x \in [-\theta, \theta]$, and 0 if $x \in [-\pi, \pi) / [-\theta, \theta]$.
> >
> > We use this observation to state partial group convolutions in terms of sampling on the entire group (Eq. 7). The fact that the distribution is defined on the entire group, is what allows us to learn the limits of the range on which the distribution is uniform.
> >
> > **[Final words]** We hope that these responses clarify your questions and concerns. We will reflect this in our manuscript by the end of this week. Please do let us know if you have any follow-up / additional questions.
> >
> > Best regards,
> >
> > The Authors

---

> > ### Comment · Reviewer_RivC · 2021-11-18
> > **Response to Authors' Comments**
> >
> > I would like to thank the authors for devoting significant efforts to rewriting the manuscript and answering my questions.
> >
> > I think the writing has been significantly improved, but there are a few additional comments:
> > - I can see that the authors have changed many of their misleading phrasings. But my previous list only contains a few examples and is not exhaustive. And there is still space for improvement. For example, "the group convolution is an operation from the group to itself" (Sec 4.2),  "considering a group convolution from a subset $\mathcal{S}^{(1)}$ to a subset $\mathcal{S}^{(2)}$" (Sec 4.2), but the group itself is not the input/output space of the convolution operation. Again, they are just examples, it might be good to go through the whole text and check them. However, I do understand that there is only limited time for making such modifications.
> > - I think the current writing structure is not ideal. Normally, you would like to get your main ideas across as soon as possible, but currently, the real content comes in on page 4. Moreover, the related works, which assume some knowledge on equivariance/invariance come before the preliminaries. In fact, I think some of the content in preliminaries can be in the appendix. Currently, it takes too long to get to your main idea.
> >
> > I am mostly satisfied with your answers to my questions. But I would like to point out that the following is still a bit strange to me: The partial conv is defined using distribution over groups. But the distribution for the discrete groups is not exactly using this definition (using a Bernoulli distribution indicating whether a group element is in or not). I understand your motivation for doing this, but this inconsistency should be resolved. I might be wrong, but maybe this fact suggests that it is worthwhile to consider a different way of defining the partial conv?

---

> > > ### Author Response · Authors · 2021-11-19
> > > **Second response Reviewer RivC**
> > >
> > > Dear Reviewer RivC,
> > >
> > > thank you very much for your response.
> > >
> > > **I can see that the authors have changed many of their misleading phrasings. But my previous list only contains a few examples and is not exhaustive. And there is still space for improvement. For example, "the group convolution is an operation from the group to itself" (Sec 4.2), "considering a group convolution from a subset  to a subset " (Sec 4.2), but the group itself is not the input/output space of the convolution operation. Again, they are just examples, it might be good to go through the whole text and check them. However, I do understand that there is only limited time for making such modifications.** Oh, that is indeed misleading. We will go through the entire text and we will filter out all misleading statements. Thank you very much for pointing this out. This will certainly help improving the quality of our work.
> > >
> > > **I think the current writing structure is not ideal. Normally, you would like to get your main ideas across as soon as possible, but currently, the real content comes in on page 4. Moreover, the related works, which assume some knowledge on equivariance/invariance come before the preliminaries. In fact, I think some of the content in preliminaries can be in the appendix. Currently, it takes too long to get to your main idea.** I understand  your concern. We have the following proposal:
> > >
> > > We propose to set the structure of the document as follows:
> > > 1. Introduction
> > > 2. Preliminaries: Group Equivariance and the Group Convolution
> > > 3. Partial Group Equivariant Networks
> > >
> > >     3.1 Partial Group Equivariance
> > >
> > >     3.2 Partial Group Convolutions
> > >
> > >     3.3 From Group Convolutions to Partial Group Convolutions
> > >
> > >     3.4 learning Group Subsets via Probability Distributions on Group Elements
> > >
> > >     3.5 Partial Group Equivariant Networks
> > >
> > > 4. Related Work
> > >
> > > 5. Experiments
> > >
> > > 6. Discussion
> > >
> > > 7. Conclusion and Future Work
> > >
> > > In other words, we propose to move the related work section after the introduction of our method. In addition, to get the idea of partial group convolution on the head of the reader as fast as possible, we propose to move the definition of the partial group convolution after the definition of partial group equivariance.
> > >
> > > Please let us know if you find this structure more coherent. We have not changed this yet, because it might make it difficult for the other reviewers to re-assess the changes of the document via pfddiff. However, if you like the idea, we promise to do so after this rebuttal period.
> > >
> > > **The partial conv is defined using distribution over groups. But the distribution for the discrete groups is not exactly using this definition (using a Bernoulli distribution indicating whether a group element is in or not). I understand motivation for doing this, but this inconsistency should be resolved. I might be wrong, but maybe this fact suggests that it is worthwhile to consider a different way of defining the partial conv?** This is a very important point. We apologize that this is not still clear in the current version of the document. We hoped that this would be clear from the independence assumption of the $\{ p_i \}$ variables, but we now realize this is actually very difficult to infer from the text.
> > >
> > > The explanation is as follows:
> > >
> > > Considering a Bernoulli distribution over each element $g_i$ denoting whether the element is sampled or not can be seen as a multinomial distribution over all possible element combinations $(g_1, g_2, ... g_n)$ denoting the probability of sampling that particular combination. As each element $g_i$ is sampled independently from one another, the probability of sampling a particular combination  $(g_1, g_2, ... g_n)$ is given by:
> > >
> > > $$p(g_1, g_2 , ... , g_n)  = \prod_{i=1}^{n}p_i$$,
> > >
> > > where $p_i$ depicts the probability of sampling each element independently. In other words, the distribution over all possible combinations can be seen as the joint distribution of the Bernoulli distributions of each element. This is because if two variables are independent, their joint distribution is given by their product, i.e., $p(X=x, Y=y) = p(X=x)P(Y=y)$ if $X, Y$ independent.
> > >
> > > We can illustrate this with a small example. Consider a group with two elements $(g_1, g_2)$. Then, if we have two Bernoulli distributions $p(g_1)=p_1$ and $p(g_2)=p_2$, the joint probability distribution is given by:
> > >
> > > $$\begin{align}
> > > p(g_1, g_2) &= p_1 p_2 \\\\
> > > p(g_1, \bar{g_2}) &= p_1 (1 - p_2) \\\\
> > > p(\bar{g_1}, g_2) &= (1-p_1) p_2 \\\\
> > > p(\bar{g_1}, \bar{g_2}) &= (1-p_1)(1-p_2) \\\\
> > > \end{align}$$
> > >
> > > Note that the sum of these probabilities is one. In other words, the resulting distribution is in fact a valid probability distribution on the group.
> > >
> > > We will make sure this is clear in the paper.
> > >
> > > Best regards,
> > > The authors.

---

> > > > ### Comment · Reviewer_RivC · 2021-11-19
> > > > **Response to Authors**
> > > >
> > > > Thanks for the explanation. While this paper still has lots of space for improvement in terms of writing, it solves an interesting problem and shows encouraging results. I have raised my score to 6, but I hope the authors could spend more time going through the text carefully.
> > > >
> > > > About the writing structure, don't trust me on this, but how about merging preliminaries into Sec 3? That is to say, you could talk about group equivariance & partial group equivariance side by side. Then you could discuss group convolution & partial group convolution side by side. In this way, you directly get to your main points after the introduction and have direct comparisons between these definitions (full vs. partial) without letting the readers go forward and backward. A minor point is that I am not sure why you need sec 3.3 as a separate part in addition to sec 3.2.
> > > >
> > > > In terms of the explanation about distributions on discrete groups, it is now very clear to me, and I am satisfied with the answer.

---

> > > > > ### Author Response · Authors · 2021-11-22
> > > > > **Response to Reviewer**
> > > > >
> > > > > Thank you very much for your comments.
> > > > >
> > > > > We have carefully gone through the entire paper and filtered out the remaining inconsistencies and misleading statements.
> > > > >
> > > > > Regarding your proposed paper structure, we have had bad experiences with such structures in previous papers, mainly because reviewers find that in that way, it is not clear what the real contribution of the paper is, and what related work / preliminaries. Therefore, we would prefer not to have that structure. **A minor point is that I am not sure why you need sec 3.3 as a separate part in addition to sec 3.2.** Good point. We can include 3.3 as a subsection of 3.2.
> > > > >
> > > > > **In terms of the explanation about distributions on discrete groups, it is now very clear to me, and I am satisfied with the answer.** We are glad to hear that our answer was clear. We have now made this point clear in the paper.
> > > > >
> > > > > Best regards,
> > > > >
> > > > > The authors

---

### Decision · Program_Chairs · 2022-01-20

**Decision:**

Reject

**Comment:**

The paper proposed learning partial and full equivariances from data in an end-to-end way. The problem studied is an important issue of existing equivariant neural networks which always assume full equivariance. However, the paper only got 3 "marginally below threshold" and 1 "marginally above threshold" even after rebuttal. The major challenges include technical parts being hard to follow due to multiple reasons, unsatisfactory paper writing (the updated version has undergone restructuring), the issue of "breaking equivariance" after multiple layers, some important experiments (such as comparing with Steerable G-CNNs) missing, etc. After rebuttal only reviewer RivC raised his/her score to "marginally above threshold". Such scores are difficult to justify acceptance. The AC appreciated the authors for making great efforts on rebuttal and revising the manuscript. However, from the review comments it is clear that the paper still needs further revision (many of those clarified for the reviewers, e.g., explaining distribution for the discrete groups and adding more experiments, should be included in the manuscript). So the AC deemed that the paper is not ready for publication.